# Advances in Lipid-Polymer Hybrid Nanoparticles: Design Strategies, Functionalization, Oncological and Non-Oncological Clinical Prospects

**DOI:** 10.3390/ph18121772

**Published:** 2025-11-21

**Authors:** Shery Jacob, Namitha Raichel Varkey, Sai H. S. Boddu, Bapi Gorain, Rekha Rao, Anroop B. Nair

**Affiliations:** 1Department of Pharmaceutical Sciences, College of Pharmacy, Gulf Medical University, Ajman 4184, United Arab Emirates; 2025mdd01@mygmu.ac.ae; 2Department of Pharmaceutical Sciences, College of Pharmacy and Health Sciences, Ajman University, Ajman 346, United Arab Emirates; s.boddu@ajman.ac.ae; 3Center of Medical and Bio-Allied Health Sciences Research, Ajman University, Ajman 346, United Arab Emirates; 4Department of Pharmaceutical Sciences and Technology, Birla Institute of Technology, Ranchi 835215, India; bapi.gn@gmail.com; 5Department of Pharmaceutical Sciences, Guru Jambheshwar University of Science and Technology, Hisar 125001, India; rekhaline@gjust.org; 6Department of Pharmaceutical Sciences, College of Clinical Pharmacy, King Faisal University, Al-Ahsa 31982, Saudi Arabia; anair@kfu.edu.sa

**Keywords:** lipid-polymer hybrid nanoparticles, oncology, types, methods, therapeutic applications, functionalization, delivery systems, patents

## Abstract

Lipid-polymer hybrid nanoparticles (LPHNPs) are the next-generation nanocarriers that integrate the mechanical strength and sustained-release capacity of polymeric cores with the biocompatibility and high drug-loading efficiency of lipid shells. Various design strategies and architectures that enhance encapsulation efficiency, stability, and targeted delivery of diverse therapeutic agents are reviewed. Commonly employed polymers, lipids, and surfactants that enable controlled drug release and enhanced pharmacokinetic performance are summarized in tabular form, while fabrication methods such as single-step, emulsification-solvent evaporation, and microfluidic techniques are discussed for their scalability and reproducibility. The therapeutic potential of LPHNPs in delivering poorly soluble drugs, phytochemicals, and genetic materials achieving synergistic therapeutic outcomes in oncological applications is comprehensively highlighted. The manuscript also includes details on ligand-based functionalization and the integration of imaging and stimuli-responsive elements to enhance targeted delivery and develop multifunctional theranostic LPHNPs systems. Furthermore, non-oncologic applications of LPHNPs in ocular, topical, and oral delivery are discussed, emphasizing their potential in treating inflammatory, infectious, and autoimmune disorders with sustained release and enhanced therapeutic efficacy. Recent patents focusing on improved biocompatibility, dual-drug encapsulation, and mRNA delivery are summarized. However, challenges such as large-scale production, reproducibility, safety, and regulatory standardization must be addressed through quality by design approaches and advanced manufacturing technologies to fully realize the clinical and commercial potential of next-generation LPHNPs.

## 1. Introduction

Nanoparticles play a transformative role in the biomedical field by enhancing the efficacy, safety, and specificity of therapeutic interventions. They achieve this by enabling the targeted delivery of a diverse range of therapeutic agents, including nucleic acids, peptides, proteins, small-molecule drugs, and diagnostic imaging probes [1]. Due to their nanoscale dimensions and modifiable surface characteristics, like size, surface charge, and functionalization of nanoparticles may enhance drug solubility and bioavailability, facilitate absorption via lymphatic pathways, enhance cellular uptake, and allow for controlled and sustained drug release. These characteristics also contribute to extended circulation time and selective accumulation at diseased sites through passive processes like the improved permeability and retention effect and active targeting with ligands like antibodies, peptides, or aptamers [2]. In addition to improving therapeutic results, this targeted approach reduces systemic side effects and off-target toxicity, two significant drawbacks of traditional formulations. Furthermore, advancements in stimuli-responsive nanoparticles have enabled on-demand drug release responding to physiological or external triggers such as pH, enzymes, redox conditions, or temperature, further enhancing precision in treatment [3,4]. Several studies have demonstrated that such nanocarrier-based delivery systems considerably improve the pharmacokinetic profiles and therapeutic indices of drugs, particularly in cancer, neurological, and infectious disease therapies.

Each class of nanocarriers offers distinct advantages but also has specific drawbacks that limit their widespread clinical translation (Table 1). Polymeric nanoparticles (e.g., polylactic-co-glycolic acid (PLGA), chitosan) often suffer from poor drug loading for hydrophilic molecules, residual solvent toxicity, and potential burst release [5]. Lipid-based nanoparticles, including liposomes, solid lipid nanoparticles, and nanostructured lipid carriers, face issues such as low physical stability, drug leakage, and limited scalability [6,7]. Inorganic nanoparticles (e.g., gold, silica, quantum dots) provide precise imaging and targeting capabilities but may exhibit long-term toxicity, poor biodegradability, and tissue accumulation [6]. Carbon nanotubes are valued for their high surface area and drug-loading capacity, but they pose serious concerns regarding cytotoxicity, immunogenicity, and environmental persistence [8]. Dendrimers, though structurally uniform and highly tunable, can be toxic at higher generations, expensive to synthesize, and difficult to scale up. Nanoemulsions and micelles are easy to formulate but may suffer from instability and premature drug release. Lipid-polymer hybrid nanoparticles (LPHNPs) are a next-generation platform that synergistically combines the robust structure and controlled release properties of polymeric nanoparticles with the biocompatibility and high drug-loading efficiency of lipid-based carriers. A recent review emphasizes the complementary benefits of LPHNPs in Alzheimer’s models, including improved BBB penetration and targeted genetic alteration [9]. It has been said that LPHNPs provide a promising platform for combining small-molecule, peptide, and nucleic acid techniques to target tau diseases, given that brain-targeting ligands and BBB-permeation procedures are included. However, despite combining the advantages of lipid and polymer-based systems, LPHNPs present challenges such as complex fabrication, batch-to-batch variability, and potential instability due to lipid-polymer interactions. These difficulties impede large-scale production, quality control, and long-term storage. Therefore, optimizing formulation parameters and developing scalable, reproducible manufacturing methods are essential. This review critically evaluates the design strategies, therapeutic applications, and translational challenges of LPHNPs to support their advancement toward clinical use.

## 2. Design and Fabrication

### 2.1. Structural Features of LPHNPs

From a structural perspective, LPHNPs can be broadly classified into four main subtypes: (a) lipid-core polymer-shell, (b) polymer-core lipid-shell, (c) self-emulsifying, and (d) matrix-structured LPHNPs. In addition, the recent emergence of cell-based biomimetic LPHNPs, which integrate biological membranes or ligands for improved targeting and immune evasion, has expanded their therapeutic potential (Figure 1). These architectures are described in greater detail in the following subsections.

#### 2.1.1. Core–Shell Architecture

LPHNPs typically exhibit a core–shell architecture that integrates the advantages of both polymeric core (e.g., PLGA, polylactic acid (PLA), or chitosan) that provides mechanical strength, protects encapsulated drugs from premature degradation, and supports sustained or stimuli-responsive drug release as well as lipid shell or monolayer (e.g., phospholipids, lecithin, or polyethylene glycol (PEG)ylated lipids) offers biocompatibility, facilitates prolonged systemic circulation, and surface functionalization with ligands for active targeting [27,28,29]. These hybrid systems are particularly attractive for targeted delivery of various therapeutics, including small molecules, recombinant proteins/peptides, small interfering RNA (siRNA) and nucleic acids [30]. LPHNPs have shown particular promise in oncology, gene delivery, and treatment of inflammatory and infectious diseases. Several examples highlight the versatility of this system: paclitaxel formulated with PLGA, stearyl amine, and soya lecithin [31]; hydroxycamptothecin (HCPT) encapsulated in PLGA with 1,2-distearoyl-sn-glycero-3-phosphoethanolamine (DSPE)-PEG2000 and lecithin [32]; baricitinib delivered using stearin-PLGA nanoparticles that demonstrated favorable physicochemical properties and facilitated drug absorption [33]; and ursolic acid incorporated into PEGylated PLGA/lipid nanoparticles that exhibited excellent stability and potent anticancer activity [34]. Similarly, chitosan-based polymer-lipid nanoparticles loaded with enoxaparin achieved 5-fold enhancement in the oral bioavailability due to enhanced intestinal absorption [35]. Collectively, the core–shell design of LPHNPs provides a versatile and robust platform capable of improving drug encapsulation efficiency, stability, and therapeutic outcomes.

#### 2.1.2. Bio-Mimetic Coatings

An advanced modification of LPHNPs involves the use of red blood cell (RBC)-camouflaged coatings to enhance in vivo stability and prolong circulation time. These biomimetic systems are developed by fusing drug-loaded polymeric nanoparticles, such as PLGA or PLA cores, with vesicles derived from purified RBC membranes [36,37]. The process involves collecting RBCs, lysing them to remove intracellular components, isolating membrane fragments, and subsequently processing them into nanoscale vesicles via sonication or extrusion. When co-incubated with polymeric nanoparticles, these vesicles spontaneously fuse to form bilayer of lipid shell surrounding a polymeric core. This biologically derived coating provides immune evasion, extends circulation half-life (t½), and reduces clearance by the mononuclear phagocyte system. In addition, the compact bilayer supports sustained drug release at a slower rate compared to conventional formulations. However, potential immunogenicity due to variability in blood group antigens poses a limitation to clinical application, necessitating donor–recipient compatibility considerations [38,39]. Advanced strategies have further functionalized RBC membranes with targeting ligands to improve specificity. For instance, RBC membranes conjugated with DSPE-PEG-folic acid enabled immune evasion and prolonged circulation when used to encapsulate nanoparticles [40]. Similarly, mannose-modified RBC membranes coated onto PLGA nanoparticles facilitated selective delivery to antigen-presenting cells in lymphoid tissues, while also protecting encapsulated antigens from rapid clearance and extending systemic retention [41]. These RBC-mimetic coatings highlight the potential of hybrid biomimetic approaches to achieve stealth properties, targeted delivery, and improved therapeutic performance of LPHNPs.

#### 2.1.3. Polymer-Caged Nanobins

Polymer-caged nanobins represent an innovative subclass of LPHNPs designed to enhance the stability, responsiveness, and specificity of liposomal drug delivery systems. These nanobins are constructed by coating conventional liposomes with pH-responsive polymer layers, typically weak polyelectrolytes such as polyacrylic acid or polymethacrylic acid, which introduce functional carboxylate groups onto the liposomal surface [42]. These groups act as anchoring sites for conjugating stimuli-responsive linkers or ligands that remain stable under physiological pH but undergo cleavage in acidic tumor microenvironments or intracellular compartments such as endosomes and lysosomes, thereby enabling controlled and site-specific drug release [43,44]. Cross-linked polymer shells improve the structural integrity of liposomes by shielding them from enzymatic degradation and serum-induced disintegration, while also enabling precise control over permeability and release kinetics [45]. This architecture supports the co-delivery of multiple drugs with distinct release profiles, facilitating synergistic therapeutic effects. Additionally, polymeric cages enhance colloidal stability by preventing aggregation and fusion during storage or circulation, which is particularly advantageous for parenteral administration [46]. Beyond stability, polymer-lipid complexes offer tunable physicochemical properties, where surface charge, hydrophilicity, and zeta potential can be optimized by altering polymer composition or crosslinking degree to prolong circulation time and minimize opsonization by the mononuclear phagocyte system. Recent advances include the development of temperature- and pH-sensitive complexes using Pluronic^®^-based hydrophilic copolymers synthesized via atom transfer radical polymerization, which demonstrated enhanced liposomal stability, effective stimuli-responsive drug release, and minimal cytotoxicity [47]. Functionalization with targeting ligands further improves site-specific delivery [48]. Moreover, novel redox-sensitive and enzyme-degradable polymer cages that disassemble in response to intracellular glutathione levels or tumor-associated proteases are being explored, offering dual or multi-stimuli responsive platforms for precision drug delivery [49].

Collectively, these nanocarriers combine the structural advantages of liposomes with the smart and customizable attributes of polymers, making them highly promising candidates for next-generation nanomedicine. The choice of fabrication method is directly influenced by the structural characteristics mentioned above, especially the core–shell arrangement, choice of lipid coatings, and polymeric architecture. This is because each technique has a different capacity to control size, shell thickness, encapsulation efficiency, and stability.

### 2.2. Materials

The rational selection of materials is central to the performance of LPHNPs, as each component contributes uniquely to particle architecture, drug loading, stability, and biological interactions. Table 2 provides various categories generally utilized in LPHNPs. Typically, biodegradable polymers form the inner matrix, providing sustained release and high drug payload capacity [50]. Surrounding lipids, including phosphatidylcholine and cholesterol, stabilize the core and enhance compatibility with biological membranes, thereby minimizing immunogenicity [51]. Utilizing low molecular weight PLGA, optimizing the polymer-to-lipid ratio, and selecting suitable solvents (e.g., dichloromethane) have led to the production of smaller and more stable particles, with high encapsulation efficiencies [52]. PEGylated lipids such as DSPE-PEG provide steric stabilization by imparting stealth properties, with PEG chain length regulating circulation t½ and reducing opsonization [53]. A growing body of literature highlights contradictory findings regarding the role of PEGylation in lipid-nanoparticles, commonly referred to as the “PEG dilemma.” While PEGylation is widely used to enhance colloidal stability, reduce opsonization, and prolong systemic circulation through steric shielding, several reports demonstrate that PEG chains can simultaneously hinder cellular uptake and endosomal escape due to reduced interactions with cell membranes and impaired fusion processes [54]. These conflicting effects have been observed across multiple nanoparticle platforms, including polymeric micelles, liposomes, and LPHNPs, where excessive PEG density or high molecular weight PEG can create a hydrophilic barrier that limits ligand-receptor binding and decreases intracellular delivery efficiency [55]. Moreover, repeated administration of PEGylated nanocarriers has been associated with accelerated blood clearance (ABC phenomenon), mediated by anti-PEG IgM production, further complicating their in vivo behavior [56]. In contrast, other studies indicate that optimized PEG architecture such as cleavable PEG linkers, low-density PEG, or pH-sensitive PEG detachment or coating with Poly (2-ethyl-2-oxazoline) can restore cellular uptake while maintaining circulation benefits [54,57]. These contradictory findings underscore the complexity of PEG engineering in LPHNP design and highlight the need for systematic evaluation of PEG chain length, density, and cleavability to balance stability with efficient intracellular delivery. Charged lipids, particularly cationic lipids such as 1,2-dioleoyl-3-trimethylammonium-propane, facilitate efficient nucleic acid binding and promote endosomal escape for effective gene delivery, whereas neutral or anionic lipids enhance biocompatibility and reduce nonspecific interaction [58]. Surfactants like poloxamers and polysorbates are often employed during fabrication to ensure colloidal stability, size uniformity, and improved encapsulation efficiency [59]. Their physicochemical attributes such as hydrophilic–lipophilic balance (HLB) and molecular structure further influence overall formulation performance. Additionally, targeting ligands can be conjugated to the nanoparticle surface to enhance receptor-mediated endocytosis, enabling tumor-specific intracellular delivery and improved outcomes in cancer chemotherapy [60]. A new biopolymer–lipid hybrid nanocarrier was developed using natural materials to improve the safety, stability, and efficiency of lipophilic drug delivery. The system features a lipid core carrying paclitaxel and a bovine serum albumin–dextran shell formed via the Maillard reaction, which enhances nanoparticle integrity [61]. Optimizing formulation parameters improved size control, stability, and drug loading capacity. The nanocarriers remained stable across different pH levels and enabled controlled, enzyme-triggered drug release.

### 2.3. Methods of Preparation

To achieve the desired physicochemical properties, appropriate preparation techniques are selected based on the structural criteria outlined before. The following section describes how certain LPHNP structures are built using nanoprecipitation, emulsification, and microfluidic techniques. A one-step technique uses methods such as nanoprecipitation, solvent diffusion, emulsification-solvent evaporation, or self-assembly to generate the lipid and polymer components in the same procedure [62]. This approach is more scalable and time-efficient, which makes it appropriate for production on an industrial scale. Recent studies have highlighted that single-step methods often result in smaller and more uniform nanoparticles with enhanced encapsulation efficiency, especially for hydrophobic drugs and nucleic acids [63]. The use of amphiphilic block copolymers and phospholipids in these methods promotes spontaneous core–shell organization driven by hydrophobic and electrostatic interactions. Furthermore, advancements in microfluidic-based synthesis have emerged as a promising variation in the single-step method, enabling precise control over nanoparticle size, polydispersity, and reproducibility through laminar flow dynamics and controlled mixing at microscale levels [64]. Collectively, these advancements position LPHNPs as a highly attractive framework for the administration of peptide and protein-based therapies, including applications in gene delivery [65]. Regardless of the method used, factors such as polymer type, lipid composition, solvent choice, and processing parameters critically influence the physicochemical characteristics and biological behavior of the resulting LPHNPs.

#### 2.3.1. Self-Assembly Nanoprecipitation

Nanoprecipitation or solvent diffusion involves dissolving the drug and polymer in a water-miscible organic solvent before adding it to an aqueous lipid solution. Hydrophobic interactions and solvent displacement drive the spontaneous assembly of a polymeric core enveloped by lipid molecules. This process promotes the formation of uniform LPHNPs, with drug entrapment efficiency enhanced by optimizing polymer-to-lipid ratios and solvent choice. The nanoprecipitation method efficiently produces LPHNPs under 100 nm, mainly for hydrophobic drugs [65]. It involves mixing a drug-loaded polymer in an organic solvent with an aqueous lipid dispersion, promoting self-assembly via hydrophobic interactions. High energy input, such as heat, stirring, or sonication-facilitates solvent removal, yielding stable LPHNPs.

#### 2.3.2. Emulsification-Solvent Evaporation

This method involves dissolving lipids, polymers, and hydrophobic drugs in a volatile organic solvent, which is subsequently emulsified in an aqueous phase under sonication or high-shear homogenization. Following emulsification, solvent evaporation drives the self-assembly of lipid layers around the polymeric core. The solvent evaporation technique is a versatile method for incorporating either hydrophobic or hydrophilic drugs into LPHNPs using single or double emulsion approaches [66]. The resulting LPHNPs are either lipid-coated polymer shell types, where a lipid layer surrounds a polymeric core, offering enhanced biocompatibility and drug release control, or polymer core-lipid shell types, where a polymer core encapsulating the drug is surrounded by a lipid shell to improve stability, targeting, and circulation time. In single emulsion, the lipophilic drug, polymer, and lipids are dissolved in a minimal volume of FDA-approved Class III organic solvents and subsequently dispersed into an aqueous phase under high-shear homogenization, followed by sonication. This results in solvent evaporation, leading to the formation of LPHNPs through lipid assembly around a polymeric core [67]. In case of double emulsion solvent evaporation, aqueous-soluble synthetic drugs or macromolecular proteins and peptides, such as insulin dissolved in an aqueous phase, are dispersed into an organic phase having the lipid and polymer [68]. This primary water-in-oil emulsion is then emulsified into an external aqueous phase to form a double or multiple water-in-oil-in-water emulsion under high-pressure homogenization, followed by sonication. Rapid solvent removal enables the formation of LPHNPs surrounded by lipid layers. The surface features of nanoparticles can be considerably influenced by process variables such as homogenization speed, sonication time, and phase volume ratio [69], as well as numerous formulation parameters [70]. Typically, higher polymer content tends to increase particle size as a result of elevated viscosity, whereas increasing the phospholipid concentration can improve encapsulation efficiency and surface charge with minimal impact on size [71]. However, excessive lipid levels may promote the formation of multilamellar structures or liposomes. Despite its versatility, the emulsification method faces several limitations. One major concern is the potential presence of residual organic solvents, which can pose toxicity risks and compromise the safety and regulatory acceptance of the final formulation, particularly for parenteral administration [72]. Additionally, high-energy input processes such as sonication and high-pressure homogenization can lead to the denaturation or aggregation of sensitive biomolecules like proteins and peptides, reducing their therapeutic potential.

#### 2.3.3. Microfluidic Technique

The method offers a nonconventional yet efficient approach for the self-assembly of LPHNPs, previously applied to both polymeric and lipid-based nanocarriers. These miniaturized systems operate within micrometer-scale channels, allowing precise fluid handling, rapid mixing, and enhanced control over particle size and yield. Due to laminar flow and low Reynolds numbers, molecular diffusion governs mixing, which is further improved using specialized geometries like microfluidic hydrodynamic focusing, Y and W type junctions, and staggered herringbone mixers [73]. Advancements in microfluidic technology have provided a powerful variation in the single-step approach. Microfluidic devices utilize laminar flow dynamics and precise mixing at the microscale to achieve reproducible nanoparticle formation with tight size control and low polydispersity [64]. By controlling flow rates and channel geometry, microfluidics allows superior encapsulation efficiency, scalability, and real-time tunability. This approach has proven particularly promising for peptide and protein-based therapeutics, as well as gene delivery applications [74]. Microfluidic techniques enable precise handling of small volumes of both miscible and immiscible fluids within microscale channels. The high surface area of these narrow channels promotes rapid and uniform mixing, allowing enhanced control over the resulting formulations [75]. Consequently, microfluidics provides a reproducible and scalable platform for producing uniform LPHNPs, supporting their industrial and clinical application [76]. Microfluidic techniques offer advantages over conventional bulk methods by enabling the production of monodisperse LPHNPs with higher encapsulation efficiency and tighter control over particle size. Control over particle uniformity is further enhanced by maintaining a balance between the polymer core formation time and the lipid coating time. Computational fluid dynamics simulations showed that homogenous nanoparticle assembly occurs only when the polymer cores τ growth surpasses the timescale needed for lipid coating [77]. In one attempt, LPHNPs were prepared via single-step nanoprecipitation and microfluidic methods, then functionalized with aptamers and antibodies [78]. Both techniques produced spherical, monodisperse particles with high entrapment efficiency (>70%). In another study, LPHNPs composed of PLGA, DC-cholesterol, and DOPE-mPEG2000 were successfully prepared using both nanoprecipitation and microfluidic methods [65]. The formulations exhibited high encapsulation efficiency, complete release within 24 h, good stability, and enhanced cellular uptake and gene silencing, demonstrating their effectiveness as oligonucleotide carriers. The comparative study highlighted microfluidics as a scalable and precise approach for producing uniform LPHNPs. It was reported that optimal nanoparticle synthesis in microfluidic hydrodynamic focusing systems depends on the interplay of flow rate, inlet design, and lipid concentration [79]. The study provides a framework for tailoring microfluidic setups to achieve precise and scalable production of therapeutic nanocarriers.

Compared to bulk mixing approaches, self-assembly nanoprecipitation typically produces LPHNPs with particle sizes of 80–200 nm and moderate encapsulation efficiencies (50–70%) [80], while emulsification-solvent evaporation improves size uniformity and drug loading, yielding particles of 100–250 nm with encapsulation efficiencies often exceeding 70% [81]. Microfluidic synthesis offers the greatest control over physicochemical attributes, frequently reducing polydispersity index (PDI) from ~0.25 to <0.10 and enhancing encapsulation efficiency by 10–20% through precise mixing and rapid supersaturation [75].

Microfluidic techniques often achieve superior siRNA encapsulation and nanoparticle uniformity compared to conventional bulk methods because they provide highly controlled and rapid mixing under laminar flow conditions, allowing instantaneous and homogeneous nanoprecipitation of lipid-polymer components [82]. This controlled environment minimizes premature siRNA leakage and promotes the formation of well-defined lipid–polymer interfaces, which is reflected in recent reports showing encapsulation efficiencies approaching 90% and markedly reduced polydispersity when microfluidic platforms are employed [83]. Comparative studies of lipid–polymer hybrid nanoparticles have consistently shown better reproducibility and stability profiles for microfluidic-produced systems than for those fabricated by bulk solvent methods [84]. Despite these clear advantages, microfluidics involves important trade-offs. The most significant limitation is low production throughput, as microfluidic devices typically require multiple parallel channels rather than simple volumetric scale-up, complicating industrial translation [85]. Additionally, microfluidic fabrication requires specialized chips, controlled pumps, and precise flow-rate optimization, and is more prone to clogging when high polymer concentrations or viscous formulations are used [86]. These operational complexities, along with device fabrication costs and sensitivity to formulation parameters, remain challenges when transitioning siRNA-loaded LPHNPs from laboratory research to large-scale manufacturing [87].

### 2.4. Quality by Design (QbD) Considerations in LPHNP Development

The application of QbD principles has become increasingly important in the systematic development of LPHNPs, as it provides a structured approach for enhancing product quality, safety, and performance. QbD implementation begins with defining the Quality Target Product Profile (QTPP), which for LPHNPs includes route of administration, required release behavior, stability profile, and therapeutic intent [88,89]. From the QTPP, critical quality attributes (CQAs) are identified such as particle size, PDI, surface charge, morphology, drug loading, encapsulation efficiency, and release kinetics as these parameters influence pharmacokinetics, cellular uptake, and biodistribution [90]. Material attributes such as polymer type and molecular weight, lipid composition, surfactant selection, and solvent system, along with process parameters like mixing rate, organic-to-aqueous phase ratio, emulsification time, and temperature, are evaluated as potential critical material attributes (CMAs) and critical process parameters [91]. Risk-assessment tools including Ishikawa diagrams, failure mode and effects analysis, and risk-ranking matrices are widely used to identify and prioritize variables that significantly affect CQAs [92]. Following risk analysis, statistical design of experiments approaches such as full factorial design, Box–Behnken design, or central composite design are applied to determine optimal formulation conditions and establish a design space that promotes robust and reproducible nanoparticle characteristics [93]. Control strategies are subsequently developed based on the defined design space, incorporating in-process monitoring, validated analytical characterization methods, and specification limits to ensure consistent LPHNP quality [94]. Overall, integrating QbD principles into LPHNP development strengthens scientific understanding, improves scalability, and aligns formulation practices with regulatory expectations for advanced nanomedicine products.

## 3. Functionalization Strategies for Targeted Therapy

Conventional anticancer therapies are limited by poor tumor targeting, rapid systemic clearance, and off-target toxicities [95]. LPHNPs can be functionalized with ligands for active targeting of tumor-specific receptors [96]. Nanoparticle size critically influences biodistribution and tumor accumulation; particles between 10 and 200 nm exhibit prolonged circulation and reduced clearance by the reticuloendothelial system compared with larger counterparts [97]. Passive targeting of LPHNPs exploits the EPR effect resulting from leaky tumor vasculature and poor lymphatic drainage, though variability of this effect among tumor types can affect therapeutic consistency [98]. Functionalization of LPHNPs with targeting ligands (folic acid, transferrin, or RGD peptides) enhances tumor specificity and intracellular drug accumulation. Ligands such as small molecules, peptides, polysaccharides, proteins, antibodies, and aptamers can be conjugated to the nanoparticle surface via electrostatic or covalent interactions to enable active targeting [99]. These ligand-modified LPHNPs undergo receptor-mediated endocytosis, achieving greater cellular uptake than non-targeted systems [100]. To avoid endosomal degradation, LPHNPs are often engineered with pH-sensitive or membrane-disruptive components that facilitate cytosolic release, allowing drugs to reach intracellular organelles and augment the overall therapeutic potential.

Surface functionalization of LPHNPs plays a critical role in determining their biological fate, and its effects extend well beyond simply increasing tumor specificity. The interaction between targeting ligands and their corresponding receptors is governed by ligand–receptor binding kinetics, including affinity (Kd) and association/dissociation rate constants (kon/koff), which collectively regulate the duration and strength of nanoparticle attachment to cell surfaces [101,102]. Although functionalization of LPHNPs with targeting ligands enhances tumor specificity and promotes intracellular drug accumulation, excessively strong ligand–receptor interactions may hinder optimal internalization or limit efficient cytosolic release [103,104]. The efficiency of targeted delivery is further influenced by receptor expression levels and recycling dynamics. Many tumor-associated receptors, such as folate receptor-α, transferrin receptor, and integrins, undergo rapid internalization followed by recycling to the plasma membrane, allowing multiple successive rounds of nanoparticle uptake and amplifying intracellular delivery [28]. Thus, the functionalization of LPHNPs modulates not only targeting specificity but also the kinetics of binding, internalization routes, intracellular trafficking, and recycling, collectively determining the overall efficiency of delivery.

### 3.1. Folate Ligands

Folate-targeted LPHNPs represent a promising strategy for tumor-specific delivery of chemotherapeutics and semisynthetic phytochemicals. By exploiting the overexpression of folate receptors on cancer cells, these systems enhance drug encapsulation, stability, and receptor-mediated uptake, resulting in improved cytotoxicity, apoptosis, and tumor inhibition compared with non-targeted formulations. Both conventional drugs such as doxorubicin [105], docetaxel [106], cisplatin [107], and mitomycin C [108] and semisynthetic phytochemicals with anticancer potential have been successfully incorporated into folate-modified LPHNPs, exhibiting favorable nanoscale dimensions, low PDI, and stable zeta potential. Preclinical evaluations have shown prolonged systemic circulation, enhanced tumor accumulation, and superior pharmacological outcomes across multiple cancer models. Nevertheless, challenges remain in achieving large-scale reproducibility, maintaining formulation stability under physiological conditions, and minimizing off-target accumulation in tissues with basal folate receptor expression. The potential of folate ligand-based active targeting was demonstrated in a folate-modified LPHNPs containing paclitaxel [109]. The system employed a self-assembly of triblock copolymer of poly(ε-caprolactone)-PEG-poly(ε-caprolactone) as the polymeric core and a lipid shell composed of lecithin and DSPE-PEG2000, with folate ligands conjugated to the surface for active targeting. The resulting LPHNPs exhibited high encapsulation efficiency, nanoscale size, enhanced uptake in folate receptor-overexpressing cancer cells, and superior tumor inhibition compared to non-targeted formulations and free paclitaxel.

### 3.2. iRGD and Tumor-Penetrating Peptides

For enhancing tumor-specific delivery and penetration of therapeutic agents, researchers have investigated cell-penetrating and tumor-homing peptides. Among them, internalizing Arg-Gly-Asp (iRGD) peptides has attracted significant attention for its dual targeting and tissue-penetration capabilities. iRGD initially binds to αβ integrins overexpressed on tumor cells and vasculature, then undergoes proteolytic cleavage to expose a CendR motif, which binds to Nrp1 and promotes tumor penetration and cellular uptake. In an earlier study, CUR-loaded LPHNPs modified with RGD peptides were formulated with emulsification-solvent volatilization method to enhance tumor targeting and assessed for their suitability for intravenous administration [110]. The optimized RGD-LPHNPs, composed of PLGA-mPEG and RGD-PEG-cholesterol copolymers with a lipid shell, exhibited high entrapment efficiency, nanoscale size, and near-neutral surface charge. Transmission electron microscopy confirmed their core–shell structure. RGD-LPHNPs outperformed free CUR in terms of cellular absorption in HUVECs, cytotoxicity, and tumor suppression in a B16 melanoma model, as evidenced by increased apoptosis and reduced angiogenesis and cell proliferation.

To improve the pharmacokinetics of the poorly absorbed natural anticancer dietary compound isoliquiritigenin, LPHNPs were modified with iRGD peptides [111]. These nanoparticles exhibited a stable structure, high drug-loading capacity, and enhanced cellular uptake through iRGD-integrin interactions. Compared with free flavonoid and non-modified nanoparticles, the modified LPHNPs demonstrated greater cytotoxicity, apoptosis, and tumor accumulation, resulting in significantly improved tumor growth inhibition in 4T1 breast cancer models.

### 3.3. Transferrin, Aptamer, and Antibody-Conjugated Systems

Alternative ligands such as transferrin, aptamers, and antibodies were investigated for cancer-targeted drug delivery beyond the folate and RGD system. Alpha-mangostin has strong anticancer potential, but its efficiency is restricted by limited solubility, low absorption, and quick clearance. Transferrin-conjugated LPHNPs improved their entrapment, sustained release, targeted uptake, and antiproliferative activity, making them a promising cancer therapy approach [112]. Recently, functionalized LPHNPs were prepared using gambogic acid to target transferrin receptors, enabling receptor-specific delivery [113]. Urolithin A was successfully encapsulated in LPHNPs by emulsion-based method, producing smaller particles with higher encapsulation efficiency than polymeric or lipid nanoparticles alone. The hybrid nanoparticle showed a lipid-rich core with this natural molecule functionalized polymer shell, enhanced cellular uptake, and strong anti-inflammatory effects by downregulating TLR4, NF-κβ, and IL-1β in a cisplatin-induced kidney injury model. These findings demonstrate the potential of LPHNPs as an effective targeted delivery system for treating chemotherapy-induced acute kidney injury.

Aptamer-functionalized LPHNPs were engineered for co-delivery of CUR and cabazitaxel in synergistic prostate cancer therapy [114]. These nanoparticles (~121 nm, +23.5 mV) enabled sustained drug release, improved cellular inhibition, and enhanced tumor accumulation, achieving significant tumor suppression in vivo at an optimal CUR: cabazitaxel ratio of 2:5. Anti-CD33 Fab′-decorated LPHNPs exhibited extended circulation in mice and improved blood levels of 1-β-D-arabinofuranosylcytosine compared with whole antibody conjugates. However, incorporating a pH-sensitive copolymer did not enhance survival in leukemic mice [115]. In another study, paclitaxel-loaded LPHNPs were functionalized with half-antibodies against carcinoembryonic antigen via postinsertion. These antibody-conjugated nanoparticles selectively targeted this blood protein-overexpressing BxPC-3 pancreatic cancer cells, enhancing cellular uptake and in vitro cytotoxicity compared with non-targeted formulations [116]. These strategies highlight the potential of ligand-based targeting to improve nanoparticle specificity and therapeutic effect.

Lung cancer remains the leading cause of cancer-related deaths, with drug resistance posing a major challenge in treating non-small cell lung cancer. Aptamer-decorated LPHNPs (~213 nm, +15.9 mV) co-loaded with a docetaxel prodrug and cisplatin showed efficient targeted uptake, strong cytotoxicity (IC_50_ ~0.71 μg/mL), and synergistic activity in vitro [117]. In diseased mice, they achieved 81.4% tumor inhibition, surpassing non-targeted or single-drug systems, highlighting their potential to overcome drug resistance and improve lung cancer therapy.

Active targeting LPHNPs require not only precise formulation of targeting ligands but also rigorous manufacturing and characterization workflows to ensure batch-to-batch consistency, safety, and clinical viability [84]. Recent advances show that microfluidic or hydrodynamic mixing methods allow tight control over nanoparticle size distribution and encapsulation efficiency, two critical quality attributes that influence biodistribution and target engagement [118]. For example, continuous lipid nanoparticle production via microfluidic mixing coupled with QbD frameworks has been shown to yield lipid nanocarriers with desired size (~70–100 nm), high nucleic acid encapsulation, and reduced heterogeneity [119]. An illustrative image (Figure 2) of functionalized LPHNPs is included to depict the role of ligand-conjugated lipid shells in receptor-mediated targeting and enhanced intracellular delivery.

## 4. Therapeutic Applications

### 4.1. Oncological

#### 4.1.1. Phytochemicals and Natural Bioactives

LPHNPs for the delivery of paclitaxel, a poorly water-soluble anticancer drug, have been described [61]. The system consisted of a lipid core derived from natural lipid mixture (beeswax and olive oil), stabilized by a polymeric shell of BSA/dextran conjugates prepared via the Maillard reaction. Optimized formulations of hybrid nanoparticles demonstrated high encapsulation efficiency (70–80%) and maintained good stability under different pH conditions, ensuring minimal drug degradation or loss. In vitro studies revealed controlled release behavior, where the polymeric shell reduced burst release and enabled both passive and enzyme-triggered drug release mechanisms. Docetaxel-loaded LPHNPs were formulated via self-assembly nanoprecipitation for controlled and sustained delivery. In breast cancer cell studies, they exhibited enhanced cytotoxicity, increased cellular uptake, lower IC_50_, and greater apoptosis than free docetaxel [120]. In vivo, LPHNPs-docetaxel achieved improved pharmacokinetics, selective tumor targeting, reduced off-target effects, and led to a marked reduction in tumor volume with better survival outcomes, confirming their potential as an effective nanocarrier for breast cancer therapy. Various phytochemicals incorporated into LPHNPs and their details are presented in Table 3.

A core–shell LPHNPs system was developed using a one-step self-assembled nanoprecipitation technique for the delivery of the highly hydrophobic alkaloid camptothecin (CPT) in colorectal cancer [125]. The designed nanoparticles (~143 nm, 91% encapsulation) displayed pH-responsive release, faster in tumor-like acidic conditions, due to the combination of polymer stability and lipid biocompatibility. In vitro and in vivo studies demonstrated enhanced cytotoxicity, cellular uptake, prolonged circulation, and targeted tumor accumulation compared to free CPT, indicating strong potential for controlled and effective cancer therapy.

A hydrophilic polymer stealth coating offers many advantages, including increased serum stability, strong membrane integrity, controlled drug release, and reduced toxicity, making it highly suitable for anticancer applications. HCPT, a water-insoluble anticancer drug, was efficiently loaded into LPHNPs by the modified emulsification-solvent evaporation method [32]. In vitro, HCPT-LPHNPs showed greater cytotoxicity and lower IC_50_ values against cancer cell lines than free HCPT, while in vivo studies demonstrated 3-fold higher drug absorption, enhanced tumor inhibition, and minimal side effects, confirming their potential as an effective antitumor delivery system. Sorafenib was encapsulated in LPHNPs made using bulk and microfluidic nanoprecipitation methods [75]. The microfluidic approach produced core–shell nanosized particles having greater drug encapsulation, controlled drug release, and Fickian diffusion behavior. Characterization confirmed drug compatibility and stability, while in vitro tests showed enhanced cytotoxicity and biocompatibility compared to free sorafenib. Formulation and optimization of resveratrol-loaded LPHNPs for the treatment of glioblastoma multiforme were reported [126]. The developed formulation showed sustained drug release (~87% over 24 h) and enhanced cellular uptake (162%), leading to superior anticancer effect compared to 5-fluorouracil. Curcumin (CUR) has limited therapeutic use due to poor solubility and short t½ [127]. Formulating CUR into poloxamer F127 nanoparticles (~49 nm) achieved over 99% encapsulation efficiency, spherical morphology, and enhanced drug release in aqueous media [128]. LPHNPs demonstrated improved immune response, strong antioxidant activity, and significant cytotoxic effects against gastric and colon cancer cells, while showing low toxicity in empty carriers, indicating their strong potential as a safe and effective delivery system for CUR. The antitumor potential of salidroside (Sal) is significantly limited by its high-water solubility, poor oral absorption, and low tumour cell uptake. This glycoside-loaded LPHNPs were developed using PLGA-PEG-PLGA copolymers and a lipid mixture (lecithin/cholesterol = 5/1, *w*/*w*) via a w/o/w double emulsification method, achieving ~65% entrapment efficiency, ~150 nm particle size, negatively charged surface (−23 mV), and sustained release without a burst effect [129]. Sal-loaded LPNPs (Sal-LPNPs) showed markedly improved anticancer activity in PANC-1 and 4T1 cell lines compared to free Sal. Both formulations inhibited cell proliferation in a dose-dependent manner, while blank LPNPs exhibited no cytotoxicity (Figure 3). Sal-LPNPs achieved significantly greater cytotoxicity at concentrations above 3.8 μg/mL, with IC_50_ values of 9.54 μg/mL (PANC-1) and 8.23 μg/mL (4T1), representing 3-fold increase in potency over free Sal underlining LPHNPs ability as an efficient delivery system for cancer therapy. This enhanced effect is attributed to the nanoparticles’ nanoscale size and lipid shell, which promote efficient cellular uptake through endocytosis.

The multiple reports have demonstrated that LPHNPs enhance cellular uptake, pharmacokinetics, tumor targeting, and cytotoxicity, while minimizing burst release and systemic toxicity. The polymeric shell allows controlled and enzyme-triggered release, leading to improved pharmacokinetics and better therapeutic outcomes. Overall, LPHNPs provide a flexible and efficient nanocarrier tool for targeted administration of hydrophobic anticancer drugs, providing improved safety, efficacy, and treatment precision.

#### 4.1.2. Co-Delivery of Drugs and Phytochemicals

LPHNPs co-loaded with drugs and phytochemicals represent a promising strategy to achieve synergistic therapeutic outcomes through combined pharmacological actions. A possible method for boosting the therapeutic potential of plant-derived bioactives is to co-load LPHNPs with many phytochemicals or combine a drug with a phytochemical. Co-encapsulation of synthetic drugs with bioactive phytochemicals, such as CUR, quercetin, or resveratrol, can improve treatment effectiveness by targeting multiple pathways, overcoming multidrug resistance, and reducing drug-induced toxicity. Similarly to nanovesicles, lipid nanoparticles, LPHNPs serve as a versatile delivery platform for the co-delivery of chemical and natural therapeutics, enabling optimized pharmacokinetics and enhanced clinical potential [130,131]. LPHNPs provide a distinct advantage in anticancer therapy by accommodating agents with diverse physicochemical characteristics through their hybrid architecture [132]. This design enables the simultaneous delivery of hydrophobic cytotoxic drugs encapsulated within the polymeric core together with charged macromolecular therapeutics including nucleic acids, oligonucleotides, siRNA, proteins, and peptides associated with the lipid shell [133]. Such dual loading capability promotes synergistic pharmacological outcomes and expands the scope of nanomedicine in cancer chemotherapy.

Coating LPHNPs with chitosan improves their stability, mucoadhesion, and cellular absorption via electrostatic interactions with minus charged membranes. CUR and paclitaxel were co-loaded into LPHNPs using nanoprecipitation, followed by surface coating with chitosan for breast cancer therapy. MTT assay studies revealed enhanced cytotoxicity, with the IC_50_ of free CUR and paclitaxel (480.06 µg/mL) reduced to 282.97 µg/mL for developed LPHNPs. In vivo pharmacokinetic studies in rats further confirmed greater drug absorption, with a 3.8-fold increase in AUC for CUR and a 6.6-fold increase for paclitaxel. These findings highlight the role of chitosan in boosting the pharmacological and pharmacokinetic performance of co-delivered CUR and paclitaxel [134].

CUR enhances the chemosensitivity of resistant cancer cells and exhibits synergistic antiproliferative effects when combined with standard chemotherapeutic agents [135]. It also induces cell cycle arrest at the G_2_/M phase, thereby increasing radiosensitivity, and promotes apoptosis through the inhibition of DNA topoisomerase II. LPHNPs were developed as an efficient co-delivery platform for lipophilic CUR and hydrophilic cisplatin to enhance cytotoxicity against ovarian cancer [136]. Utilizing chitosan and Lipoid S75, nanoparticles were synthesized by the ionic gelation method, forming cationic particles (~225 nm) with >80% drug entrapment and regulated dual-drug release without burst effect. In vitro studies using A2780 ovarian cancer cells and 3D tumor spheroids demonstrated improved cellular uptake, chemosensitization, and synergistic cytotoxicity of the co-loaded system compared to single-drug formulations (Figure 4). These findings indicate that LPHNPs offer a promising nanocarrier platform for the simultaneous delivery of drugs and phytochemicals in cancer therapy.

A LPHNPs system was developed for the co-delivery of docetaxel and CUR to improve pharmacological outcomes against metastatic castration-resistant prostate cancer [137]. The optimized formulations exhibited nano size (169.6 nm) with a positive charge of 35.7 mV, high drug entrapment, and controlled delivery behavior. In vitro studies on PC-3 human prostate carcinoma cells demonstrated significantly enhanced cytotoxicity and synergistic activity compared to single-drug formulations. Furthermore, in PC-3 tumor xenograft-bearing mice, the developed formulation showed superior tumor growth inhibition with minimal systemic toxicity. These findings highlight the potential of dual-drug-loaded LPHNPs as an effective nanomedicine platform for synergistic prostate cancer therapy.

A LPHNPs system co-encapsulating cisplatin and CUR was developed for the targeted treatment of cervical cancer [138]. The formulation exhibited synergistic cytotoxicity and enhanced antiproliferative activity against HeLa and HUVEC cell lines, surpassing the efficiency of single-drug-loaded nanoparticles. The dual mechanism involved drug-mediated DNA crosslinking and apoptosis, complemented by CUR’s antioxidant and anti-inflammatory effects, which sensitized tumor cells to chemotherapy. In vivo studies in BALB/c mice bearing cervical tumors confirmed superior antitumor effect, attributed to enhanced permeability and retention (EPR) effect-based tumor targeting and synergistic action of both agents.

Chrysin is a natural flavonoid known for its strong antioxidant, anti-inflammatory, anticancer, and neuroprotective properties [139]. Bioenhancers with medicinal qualities, like piperine, have shown promise in the treatment of disorders like diabetes, cancer, arthritis, and cardiovascular diseases [140]. By overcoming resistance mechanisms and increasing the effect of anticancer medications, herbal bioenhancers show promise in the fight against complex disorders like cancer [141]. Chrysin-piperine-loaded chitosan-lecithin nanoparticles demonstrated high encapsulation efficiencies (90.5% for piperine and 79.2% for chrysin) [142]. The formulation exhibited selective cytotoxicity, significantly inhibiting PANC cancer cell proliferation (IC_50_ = 14 µg/mL) while sparing normal HFF cells (IC_50_ > 500 µg/mL). qPCR and fluorescence analyses confirmed apoptosis induction through the activation of caspase-3, -8, and -9, accompanied by upregulated IL-6 expression. Additionally, the nanoparticles displayed strong antioxidant activity in various assays.

#### 4.1.3. Gene and siRNA

siRNA is a promising therapeutic tool that works by silencing specific genes through the RNA interference mechanism, leading to the degradation of target messenger RNA (mRNA). This strategy has wide-ranging clinical applications [143]. In oncology, siRNA can suppress oncogenes such as KRAS, BCL2, and VEGF, thereby reducing tumor growth, angiogenesis, and metastasis. Overall, siRNA therapies offer high specificity, versatility, and potential for personalized medicine, with several approved drugs already in use and many others advancing through clinical trials. Cationic LPHNPs represent a new platform that integrates the favorable features of both polymers and lipids with greater biocompatibility and transfection efficiency (cationic lipids) for siRNA delivery [85]. The LPHNPs architecture offers several advantages: the polymeric component imparts improved stability and reduced cytotoxicity compared to purely cationic lipid nanoparticles, while the lipid layer ensures high siRNA encapsulation efficiency through electrostatic interactions [133]. PLA-PEG-PLA triblock copolymer was synthesized by ring-opening polymerization of lactide using PEG as a macroinitiator, and subsequently hybridized with cationic lipids via sonication to form LPHNPs [144]. Various characterizations confirmed the nanoparticles’ size distribution and morphology, while the MTT assay demonstrated improved biocompatibility compared to pure lipid nanoparticles. The fabricated LPHNPs achieved high siRNA encapsulation (~80%) and successfully delivered IGF-1R-siRNA into MCF7 breast cancer cells. Cellular uptake studies using flow cytometry and fluorescence microscopy confirmed efficient internalization, and RT-PCR analysis revealed significant gene silencing with 70% downregulation of IGF-1R expression.

LPHNPs with a PLGA core and lecithin/PEG-lipid shell were prepared by the double solvent evaporation method to encapsulate siRNA to silence prohibitin 1, a protein upregulated in non-small cell lung cancer [145]. These nanoparticles achieved a 100-fold higher area under the curve (AUC) and prolonged circulation (t½ = 8 h) in mice compared to naked siRNA (cleared in 30 min). In vitro, they produced 90% silencing, significantly inhibited NCI-H460 cell proliferation, and increased apoptosis 5-fold. In vivo, they reduced tumor weight by 50% in xenograft models. Similarly, fluorodendrimer-incorporated LPHNPs with PLGA core and DSPE-PEG shell were engineered for siRNA delivery, with ~90 nm particles effectively inhibiting A549 cell growth and yielding an apoptotic rate of 19% versus 9% in controls, confirming their efficiency for siRNA therapy [146]. LPHNPs made of lipidoid L5N12 and PLGA improve siRNA delivery by reducing surface charge and toxicity compared to traditional cationic lipids [147]. L5N12 enables efficient siRNA binding while offering key advantages such as lower dose requirements, absence of permanent positive charge, and reduced reactive oxygen species related toxicity [148]. In this study, LPHNPs were fabricated using a cost-effective thiol-ene microfluidic chip, in which TNF-α-targeting siRNA in sodium acetate buffer and an organic phase containing L5N12 and PLGA were co-injected under controlled flow conditions to produce particles of ~70, 110, and 220 nm. The resulting dispersions were purified by solvent evaporation and centrifugal filtration, yielding size-controlled, siRNA-loaded LPHNPs with high encapsulation efficiency for gene-silencing applications. Pulmonary delivery studies with SPECT/CT imaging revealed similar clearance profiles across all formulations after intratracheal administration, with minimal off-target accumulation in the liver.

Preclinical studies in triple-negative breast cancer (TNBC) models indicate that ligand-directed, multi-siRNA LPHNPs improve intracellular delivery and broaden the therapeutic index by targeting tumor-associated receptors while concurrently silencing multiple signaling pathways. Optimization of ligand presentation through site-specific conjugation, PEG spacer tuning, and avidity mapping can balance receptor binding with tissue penetration [149]. Endosomal escape remains a central challenge, requiring the incorporation of ionizable lipids with pKa values between 6 and 7, pH-sensitive motifs, or membrane-active peptides, alongside quantitative assays such as Gal8 recruitment to monitor escape efficiency while minimizing cytotoxicity [150]. Antibody-functionalized LPHNPs were investigated as modular carriers that specifically engage tumor-associated receptors such as epidermal growth factor receptor and tumor-associated calcium signal transducer 2, both of which are frequently overexpressed in aggressive subtypes like TNBC [151]. Incorporation of multiplexed siRNA payloads enabled simultaneous multi-check point knockdown, thereby reducing reliance on a single target and potentially limiting off-target effects and adaptive resistance. In TNBC cell models, actively targeted multi-siRNA LPHNPs demonstrated enhanced intracellular delivery and an improved therapeutic index compared to non-targeted controls.

Recent advances show that LPHNPs are versatile vectors that may effectively translate mRNA therapy in pulmonary and immune cells for cancer immunotherapy while also overcoming the biological barriers to mRNA therapy [152]. A proof-of-concept study demonstrated that pH-responsive polymer cores encased in lipid shells allow for cytosolic transport of nucleic acids and mRNA by stimulating endosomal rupture, while the lipid exterior promotes cellular uptake and reduces systemic reactogenicity. This design has successfully transfected dendritic cells and pulmonary endothelial cells in vivo [153].

#### 4.1.4. Pulmonary

LPHNPs overcome limitations of conventional inhalation therapies, such as rapid clearance, poor solubility, and low deposition efficiency in the lungs. Their nanoscale size enables deep lung penetration, sustained release, and enhance pharmacokinetic performance, making them particularly attractive for the treatment of respiratory infections, inflammatory lung diseases, and pulmonary cancers. An innovative LPHNPs-based inhalable formulation was developed for roflumilast, aimed at targeted release to alveolar macrophages, the key regulators of inflammation in pulmonary disease [154]. These hybrid systems were engineered to target the CD206 receptor. The developed mannose-functionalized hybrid fluorescent nanoparticles (Man-LPHFNPs@Roflumilast) displayed nanoscale size, negative zeta potential, core–shell morphology, sustained drug release, and dense PEG surface that improved mucus penetration. They exhibited excellent cytocompatibility with bronchial epithelial cells and macrophages, with the latter showing active uptake via mannose-mediated endocytosis. For pulmonary administration, a nano-into-micro strategy was employed, encapsulating the nanoparticles into polyvinyl alcohol/leucine-based microparticles using spray drying, yielding a stable dry powder formulation optimized for inhalation therapy. However, in cystic fibrosis, PEGylation did not enhance mucus penetration, while non-PEGylated LPHNPs demonstrated superior diffusion, cellular uptake, and gene silencing, making them more suitable for local siRNA delivery [155].

Ivermectin is being investigated for cancer therapy but has limited solubility in water and low absorption [156]. To overcome these challenges, ivermectin-loaded LPHNPs were formulated for potential pulmonary delivery. The design incorporated polycaprolactone (polymeric core), lecithin (lipid shell) and Pluronic F127 (external stabilizer). The nanoparticles were spherical (302–350 nm), with 68–80% entrapment efficiency and sustained release of 50–60% over 96 h. Solid-state analyses confirmed structural stability, while flowability indices indicated good handling properties. Aerosolization studies using a twin-stage impinger showed fine particle fractions of 18.53–24.77%, supporting the feasibility of pulmonary administration. Overall, these findings suggest the developed LPHNPs as a promising inhalable system for enhancing anticancer effects.

Gentamicin and CUR-loaded LPHNPs were developed with uniform size (~340 nm), high encapsulation (70%), and sustained drug release [157]. These nanoparticles effectively killed planktonic and biofilm-associated bacteria, disrupted established biofilms, and were safely internalized by macrophages to eliminate intracellular bacteria.

Dry powder inhaler formulations of siRNA-loaded LPHNPs were optimized for aerosolization, and their interactions with pulmonary surfactant were studied using surface-sensitive techniques [158]. Findings showed that while pulmonary surfactant organization (single bilayers vs. multilayers) did not affect binding affinity, the physicochemical properties of LPHNPs, particularly membrane fluidity and electrostatic interactions, governed retention, translocation, and interaction pathways. Pathological microenvironments further promoted LPHNPs deposition, offering important insights for designing such formulation to enhance siRNA transport across the air-blood barrier. Building on these findings, further investigations examined how siRNA-loaded LPHNPs, designed for treating lung inflammation, interact with pulmonary surfactant and influence its physiological function [159]. In vitro biophysical analyses revealed that LPHNPs exhibit intrinsic surface activity, forming interfacial films that collapse at 49 mN/m and transiently compete with surfactant components at the air-liquid interface. At higher surface pressures, they are displaced into the aqueous subphase, enabling passage across the pulmonary surfactant barrier. While LPHNPs influence the organization and mixing of pulmonary surfactant domains, they do not compromise its essential biophysical function under physiological conditions.

### 4.2. Non-Oncological

In recent years, researchers have explored their role in antimicrobial and antifungal drug delivery, ocular therapeutics, central nervous system targeting, vaccine and immunotherapy development, as well as treatments for inflammatory and autoimmune disorders. These applications are still emerging and less explored compared to cancer therapies, but growing evidence highlights the promise of LPHNPs in facilitating drug absorption, reducing systemic toxicity, enabling targeted delivery, and supporting combination approaches that integrate therapy with controlled release and, in some cases, diagnostics.

#### 4.2.1. Ocular

Both lipid-based systems and polymeric nanoparticles have several advantages in ocular drug delivery, including minimal ocular irritation, yet have limited corneal permeation, short residence time, and leakage issues. To address the major drawbacks of individual formulations, LPHNPs were developed. In case of uveitis, difluprednate-loaded LPHNPs were developed using a PLGA polymeric core encapsulating drug and a lipid shell for stability [160]. The optimized system exhibited nanoscale size (~117 nm), high drug entrapment (>92%), and good physicochemical properties ideal for ophthalmic use. Studies confirmed their core–shell morphology, sustained release in simulated tear fluid, and nearly 4-fold higher corneal permeation compared to free drug solution. Ex vivo imaging demonstrated stromal penetration, while histopathology and HET-CAM assays verified ocular safety, highlighting the potential of the developed formulation in uveitis therapy.

Core–shell LPHNPs have been shown to rapidly reach human conjunctival epithelial cells within 30 min of administration [161]. Cationic chitosan LPHNPs have also been reported to enhance ocular gene delivery by facilitating endolysosomal escape and improving transfection efficiency [162]. It was reported that hyaluronic acid-modified LPHNPs can effectively target CD44 receptors on retinal pigment epithelium following vitreous injection [163]. A beneficial effect of this finding is that targeting CD44 receptors enhances site-specific drug delivery to the retinal pigment epithelium, potentially improving treatment effectiveness while minimizing off-target effects. Hyaluronic acid-modified LPHNPs were developed as nanocarriers for moxifloxacin to overcome its poor ocular penetration [164]. In vivo rabbit studies revealed markedly enhanced drug retention, with mean residence time and AUC_0–6_ h increasing 6.74 and 4.29-fold compared to a commercial product. In vitro studies further showed a 3.29-fold rise in corneal permeability, attributed to HA-mediated receptor-driven endocytosis. Moreover, hyaluronic acid modification enables targeted interaction with CD44 receptors on retinal pigment epithelium, supporting its potential for improved ocular drug delivery.

LPHNPs show strong potential as carriers for siRNA and antisense oligonucleotides in ocular therapy. They enhance stability, cellular uptake, and gene silencing, making them valuable for conditions such as age-related macular degeneration, diabetic retinopathy, glaucoma, and corneal neovascularization [165]. While promising, research and clinical translation in this area are still limited. Taken together, these findings highlight the versatility of LPHNPs as next-generation ocular drug delivery systems.

#### 4.2.2. Topical

Topical drug delivery faces major challenges such as poor solubility, limited skin penetration, and rapid drug clearance, which often reduce therapeutic potential. In topical application, LPHNPs offer a dual advantage by combining controlled drug release with limited transdermal absorption. These characteristics make them particularly attractive for localized treatment of infections, inflammatory conditions, and wound healing. LHNPs are valuable carriers for topical antibiotics, enabling controlled and sustained drug delivery at infection sites, minimizing the need for frequent application, and enhancing patient compliance. LPHNPs of the antibacterial drug norfloxacin, which has low lipophilicity, were prepared using PLA by the emulsification-solvent evaporation technique [166]. The developed nanoparticles showed particle sizes of 178–221 nm, narrow PDI (0.206–0.383), and positive zeta potential (+23 to +41 mV). The formulation exhibited strong antimicrobial activity against *Staphylococcus aureus* and *Pseudomonas aeruginosa*, released ~90% of the drug within 24 h, and remained stable at refrigerated storage. In another study, norfloxacin-LPHNPs were prepared using Precirol ATO and Eudragit RL100 and incorporated into an HPMC-based topical gel [167]. The optimized formulation exhibited a particle size of 159 nm, positive zeta potential, 92.61% drug release, and 79.2% entrapment efficiency. The gel demonstrated sustained release over 24 h and showed superior antibacterial activity against *Staphylococcus aureus*, *Acinetobacter baumannii*, and *Pseudomonas aeruginosa* compared to conventional drug-containing gel.

Hesperidin is a bioflavonoid with strong wound-healing activity but limited topical bioavailability. To overcome this, hesperidin-loaded LPHNPs were developed using a double emulsion solvent evaporation method and optimized with Box–Behnken response surface methodology [168]. The optimized formulation (HLPHN4) exhibited particle size of 91.43 nm, zeta potential of +23 mV, 79.97% drug release, and 92.8% encapsulation. In vitro studies demonstrated sustained drug release over 24 h along with potent antioxidant activity in the DPPH assay.

LPHNPs-loaded with antimicrobial agents in a topical gel represent a promising option for treating resistant bacterial burn wounds. LPHNPs-loaded with fusidic acid were employed as a novel approach for treating resistant bacteria in burn wound infections [169]. The nanoparticles (~311 nm, +24 mV, 79% entrapment) were incorporated into a carbopol gel, which showed superior skin permeation and retention compared to conventional formulations. In an MRSA-infected murine burn model, the drug containing LPHNPs gel significantly reduced bacterial load, accelerated wound contraction, and improved histopathological outcomes.

An investigation explored the repurposing of dapsone for acne treatment by incorporating it into LPHNPs and formulating a topical carbopol gel [170]. The optimized HN-4 formulation showed nanosize (277 nm), good entrapment (75.81%), and spherical morphology. Gel formulation demonstrated suitable physicochemical properties, sustained skin permeation, antimicrobial activity, and stability, while being non-irritant in HET-CAM testing.

The successful development of LPHNPs for hydrocortisone demonstrated improved skin delivery compared with simple drug solution [171]. Histological analysis confirmed safe epidermal thickening, and in vivo testing in a croton oil-induced ear rosacea model revealed strong anti-inflammatory activity.

Capsaicin and anti-TNFα siRNA were co-delivered using cyclic cationic LPHNPs for chronic skin inflammation [172]. The optimized system showed nanosized particles with high entrapment efficiency and deep skin penetration. In an imiquimod-induced psoriatic model, the dual-loaded LPHNPs markedly downregulated TNFα, NF-κB, IL-17, IL-23, and Ki-67 expression, outperforming single-drug formulations and showing results comparable to Topgraf^®^. Overall, LPHNPs represent a versatile and effective platform for topical drug delivery, offering the potential to overcome conventional barriers and achieve enhanced therapeutic outcomes in dermatological applications.

#### 4.2.3. Oral and Nanophytomedicine

The oral route is the most convenient and patient-compliant mode of drug delivery; however, poor solubility and limited absorption often restrict the clinical utility of several bioactive compounds. Sesamol, a natural antioxidant and anticancer agent, has limited therapeutic potential due to poor solubility and low bioavailability. To address this, drug-loaded LPNPs were developed and optimized using homogenization and ionotropic gelation [173]. The optimized formulation showed nanosized, stable, spherical particles with efficient drug encapsulation and sustained release over 24 h. Compared to pure sesamol, the developed drug-loaded LPNPs exhibited greater intestinal permeation, enhanced antioxidant and anticancer activity, and a 3.33-fold improvement in drug absorption, indicating their effectiveness as an oral delivery system for sesamol.

The bioavailability of orally administered agents, especially anticancer drugs often exhibits significant interpatient variability, which can have a greater clinical impact compared to intravenous formulations. The formulation difficulty of fisetin arises mainly from its poor aqueous solubility, low oral absorption, and rapid metabolism, which limit its pharmacological outcomes [174]. Additionally, fisetin is prone to instability under physiological conditions and undergoes extensive first-pass metabolism, reducing its systemic availability. Fisetin-loaded LPHNPs were developed to improve its efficiency against severe acute pancreatitis [175]. The optimized formulation showed high entrapment, strong mucoadhesion, nanoscale size, and sustained release. In vivo, pretreatment with drug-loaded LPHNPs protected rats from inflammed pancreas and multi-organ injury more effectively than free drug or controls, highlighting their potential as a nanophytomedicine for pancreatitis management. Optimizing the balance between mucoadhesion and mucus penetration is crucial for effective intestinal absorption of LPHNPs. These nanoparticles are absorbed via paracellular, transcellular, lymphatic, or receptor-mediated pathways, each influenced by their surface properties and modifications. Figure 5 shows a schematic representation of the intestinal absorption pathways of LPHNPs. Table 4 presents a comparative quantitative overview of diverse LPNP systems, highlighting key therapeutic performance indicators such as IC_50_ reduction, bioavailability enhancement, tumor growth inhibition, and in vivo gene-silencing efficiency. By summarizing outcomes across different drug classes, polymer–lipid compositions, and disease models, this table provides a clear depiction of how LPHNP formulations improve pharmacological response and therapeutic index across oncology and gene-delivery applications.

#### 4.2.4. Theranostics and Imaging

In recent years, LPHNPs have gained attraction as stable, biocompatible carriers for imaging agents such as quantum dots, gold/inorganic nanocrystals, and MRI/CT contrast moieties [177]. These hybrid systems can encapsulate or conjugate diagnostic payloads, allowing them to serve both therapeutic and imaging (theranostic) functions. Recent work has extended these strategies: advanced contrast agents with extended circulation and improved via dual/multimodal contrast materials are being integrated with LPHNPs-type or related platforms for more accurate tumor detection and imaging [81]. Additionally, multimodal approaches integrate radioisotopes, photoacoustic probes, CT agents, and upconversion nanoparticles, expanding diagnostic capabilities within a single nanoplatform [178]. Fluorescent dyes, especially new NIR-II variants, enable deeper tissue penetration with improved brightness and stability [179]. Quantum dots, including Cd-based and biocompatible types, provide high brightness and narrow emission spectra, with recent advances focusing on surface modification and integration into hybrid matrices for safer and more stable imaging [180]. Superparamagnetic iron oxide nanoparticles remain key for MRI contrast, magnetic particle imaging, and hyperthermia, and are effectively incorporated into LPHNPs for multifunctional theranostics [181].

Another strategy was reported to integrate diagnostic functionalities into LPHNPs synthesized via the nanoprecipitation technique [182]. In this approach, gold nanocrystals and quantum dots were covalently conjugated to PLGA chains through esterification reactions, thereby imparting optical and imaging capabilities to the polymeric core. The resulting hybrid system was subsequently self-assembled with soybean lecithin, forming a stabilizing lipid monolayer, and PEGylated to enhance steric stabilization and prolong systemic circulation. In vitro studies using the murine macrophage cell line J774A.1 demonstrated efficient cellular uptake and retention of these functionalized LPHNPs, highlighting their potential as dual-modality imaging probes. The incorporation of gold nanocrystals enables surface plasmon resonance-based contrast for optical imaging, while quantum dots provide highly photostable and tunable fluorescence signals.

Some LPHNPs are stimuli-responsive: e.g., pH, redox, enzyme, or temperature sensitive. Upon reaching the tumor microenvironment or intracellular compartments (e.g., endosomes), those triggers help to release therapeutic cargo. The imaging part can report on localization, accumulation, or sometimes even the trigger event [28]. A novel reactive oxygen species-responsive, size/shape-transformable LPHNPs was engineered to co-deliver paclitaxel and the collagen-inhibitor losartan for improved tumor penetration. The system was based on a PLA-thioketal-PEG copolymer, with losartan-loaded micelles forming the core and paclitaxel-loaded liposomes as the shell. Upon reactive oxygen species exposure, spherical LPHNPs (~120 nm) transformed into smaller discoid nanoparticles (~40 nm), enhancing tumor infiltration. In vitro studies in 3D tumor spheroids and in vivo experiments in 4T1 tumor-bearing mice demonstrated that the transformable LPHNPs exhibited superior penetration, effective collagen modulation, and significantly greater antitumor effect compared to non-transformable systems [183].

LPHNPs system incorporating magnetic Fe_3_O_4_ beads was developed for stimuli-responsive, radio frequency-triggered drug release [184]. The nanoparticles exhibited long-term stability in phosphate buffer, with controlled loading of CPT and Fe_3_O_4_. Application of a remote radio frequency magnetic field enabled on-demand CPT release and enhanced uptake by MT2 mouse breast cancer cells, resulting in significant in vitro growth inhibition. The platform’s simple preparation, stability, and controllable release highlight its potential to improve cancer chemotherapy outcomes. A representative schematic of theranostic LPHNPs is provided to illustrate their integrated therapeutic and imaging capabilities (Figure 6).

## 5. Translation Challenges

### 5.1. Patents and Clinical Trials

In the past five years (2020–2024), several patents have been filed describing novel formulations and applications of LPHNPs, underscoring the growing research interest in these systems for advanced drug delivery (Table 5). The inventions primarily focus on improving biocompatibility, targeted delivery, and dual-drug encapsulation capabilities. Recent patents (e.g., WO/2024/252406, 202410758382.8) disclose hybrid systems combining biodegradable polymeric cores (such as PLGA or poly(β-amino ester)) with lipid shells (e.g., lecithin), designed to enhance stability, cellular uptake, and mRNA delivery efficiency. Other patents explore dual-targeting mechanisms for cancer therapy (17228224), co-delivery of chemotherapeutics such as vincristine sulfate and lomustine (2021101545), and formulations for myocardial infarction treatment employing amlodipine-loaded LPHNPs (202341017235). Notably, some applications expand beyond oncology into immunologic adjuvant development (202211651315.3) and non-viral gene delivery, demonstrating the platform’s versatility. These innovations emphasize controlled release profiles, enhanced pharmacokinetics, and enhanced immune response compared to conventional nanoparticles. Despite extensive preclinical research demonstrating the potential of LPHNPs for drug delivery, there are currently no registered clinical trials evaluating canonical LPHNPs systems in humans. Specifically, formulations composed of a polymer core (such as PLGA) surrounded by a lipid shell (e.g., lecithin or DSPE-PEG) have not yet progressed to clinical evaluation.

Despite their demonstrated advantages in stability, drug-loading capacity, and targeting efficiency, the hybrid architecture of LPHNPs complicates large-scale production and leads to batch-to-batch variability, a significant barrier for regulatory approval [87]. Adding to this complexity, regulatory agencies lack clear classification frameworks for hybrid nanostructures, making it difficult to establish standardized characterization methods and critical quality attributes. Safety and immunogenicity concerns persist as well, with insufficient long-term toxicity, biodistribution, and clearance data for several canonical lipid and polymer components. Furthermore, scale-up challenges remain unresolved, as laboratory techniques such as nanoprecipitation and sonication do not easily translate to GMP-compliant manufacturing platforms [185]. The translational gap is further widened by the limited predictive value of current preclinical models, which often fail to replicate human pharmacokinetics and immune responses. Finally, intellectual property constraints and limited commercial incentives arising from complex, often proprietary formulation components also hinder industrial investment and slow the transition of LPHNPs toward clinical evaluation.

### 5.2. Future Perspectives

Although LPHNPs have demonstrated remarkable versatility and efficiency as multifunctional nanocarriers, several scientific and translational challenges continue to limit their large-scale commercialization and clinical deployment. These challenges span manufacturing scale-up, safety and immunogenicity, regulatory frameworks, and the integration of smart and personalized features for next-generation therapeutics. Scaling up LPHNPs is difficult because maintaining consistent particle size, zeta potential, and drug loading is challenging during large-scale production. A critical issue is batch-to-batch reproducibility, as small variations in polymer molecular weight, lipid composition, solvent evaporation rates, or mixing conditions can markedly influence particle size, polydispersity, drug loading, and release kinetics, making regulatory approval difficult [84]. Advanced methods such as high-shear homogenization and microfluidics show promise, but technical, economic, and regulatory barriers still limit their industrial adoption [185,186]. Storage stability also remains a major concern; hybrid formulations are prone to aggregation, lipid phase transitions, and hydrolytic or oxidative degradation during long-term storage, especially under non-lyophilized conditions [28]. Another important limitation is immunotoxicity, since cationic lipids and certain polymeric components may trigger complement activation, cytokine release, or rapid clearance by the mononuclear phagocyte system, necessitating extensive safety profiling [187]. The lack of standardized toxicology tests causes inconsistent safety data, emphasizing the need for biodegradable materials, surface modifications like PEGylation, and comprehensive in vivo safety studies for clinical translation [188]. From an economic standpoint, cost-effectiveness analysis is essential because LPHNPs require high-purity lipids and biodegradable polymers, multi-step synthesis, and specialized characterization techniques, all of which increase manufacturing cost compared to conventional nanocarriers [189]. Furthermore, patient stratification/personalized nanomedicine is increasingly recognized as a prerequisite for successful clinical translation, as inter-patient variability in immune responses, tumor microenvironment, enzyme expression, and biodistribution patterns can markedly affect therapeutic outcomes [190]. The regulatory framework for hybrid nanomedicine like LPHNPs is still unclear, as no global guidelines specifically address their dual polymer–lipid nature. Existing regulations adapted from polymeric or lipid nanoparticle standards often overlook their complex interactions. Applying QBD principles is crucial for ensuring consistent quality and safety, requiring detailed documentation of design space, risk assessment, process monitoring, and validated analytical methods. Lipid nanoparticles have become pivotal in advancing modern therapeutics, exemplified by their role in mRNA COVID-19 vaccines [191]. Beyond this, they show significant potential for treating various conditions, including genetic disorders, cancers, metabolic, and infectious diseases [192,193,194].

LPHNPs have gained considerable attention due to their structural stability and efficient dual-phase encapsulation; however, recent studies emphasize the need for comparative evaluation with other emerging hybrid nanocarrier systems. For instance, lipid–inorganic hybrids such as lipid-coated gold nanoparticles and silica–lipid composites demonstrate enhanced imaging capabilities and photothermal effects that LPHNPs alone cannot inherently provide [195]. Similarly, biomimetic nanocarriers including cell membrane-coated nanoparticles and extracellular vesicle inspired hybrids offer superior immune evasion and prolonged circulation relative to conventional LPHNPs [196]. Polymer–drug conjugate micelles, polymer–lipid micelleplexes and smart polymeric micelles have also shown improved stability and tunable release kinetics, presenting unique advantages over LPHNP systems in gene and combination therapy [197,198]. Despite these advances, few reviews offer a systematic comparison of these platforms, resulting in an incomplete understanding of where LPHNPs outperform competing hybrid systems and where limitations remain. Incorporating such comparative analysis is essential to contextualize the evolving role of LPHNPs in nanomedicine and to guide the rational design of next-generation hybrid systems.

## 6. Conclusions

LPHNPs have evolved into a powerful and versatile platform that bridges the gap between polymeric and lipid-based delivery systems. Their ability to combine mechanical stability, controlled release, and biocompatibility has positioned them as promising candidates for precision drug delivery, gene therapy, and theranostic applications. Continued advancements in formulation design, surface engineering, and functionalization have enabled improved targeting, responsiveness, and therapeutic potential across a wide range of diseases. However, realizing their full clinical and commercial potential will depend on overcoming key translational hurdles through standardized manufacturing, robust safety evaluation, and well-defined regulatory frameworks. Several key research gaps continue to hinder the clinical translation of LPNPs. Beyond basic optimization of size, charge, and loading, deeper understanding of ligand–receptor kinetics, intracellular trafficking, and receptor recycling is still lacking. Although microfluidic technologies improve reproducibility, their scalability, cost, and regulatory suitability remain underexplored. The long-term immunological behavior of LPHNPs, including biodistribution and responses to repeated dosing, is also insufficiently characterized. Conflicting findings regarding PEGylation highlight the need to evaluate alternative stealth strategies such as cleavable PEG, zwitterionic coatings, and poly(2-oxazoline) lipids. Furthermore, challenges such as poor endosomal escape for nucleic acid delivery, absence of standardized animal models, and limited correlations between formulation parameters and therapeutic outcomes require focused investigation. Addressing these gaps is crucial for advancing LPHNPs toward reliable and clinically viable nanomedicine applications. Collaborative efforts among researchers, industry, and regulatory agencies will be critical in transforming LPHNPs from experimental systems into clinically approved nanomedicine platforms that redefine the future of targeted and personalized therapy.

## Figures and Tables

**Figure 1 pharmaceuticals-18-01772-f001:**
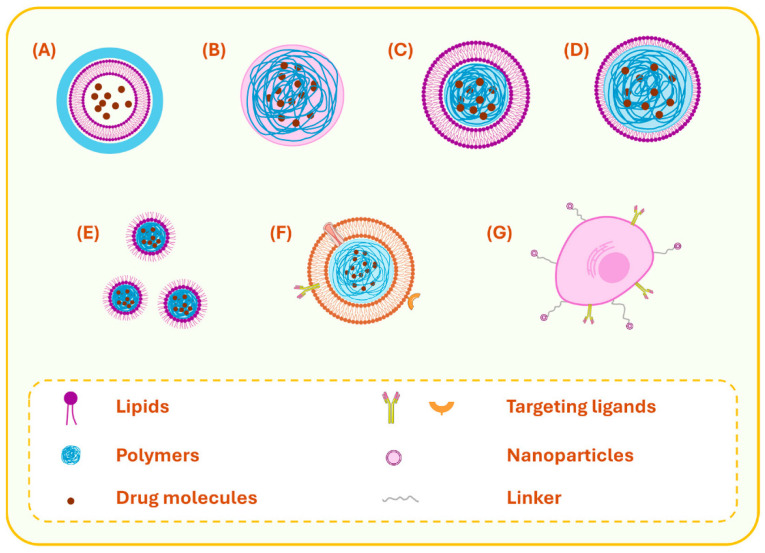
Various types of lipid-polymer hybrid nanoparticles (LPHNPs): (**A**) lipid-core polymer-shell type; (**B**) matrix-structured LPHNPs; (**C**,**D**) polymer-core lipid-shell type; (**E**) self-emulsifying LPHNPs; (**F**) cell membrane–biofunctionalized nanoparticles; and (**G**) cell-polymeric nanoparticle hybrid vectors. This figure is adapted from [26], used under a CC BY 4.0 license.

**Figure 2 pharmaceuticals-18-01772-f002:**
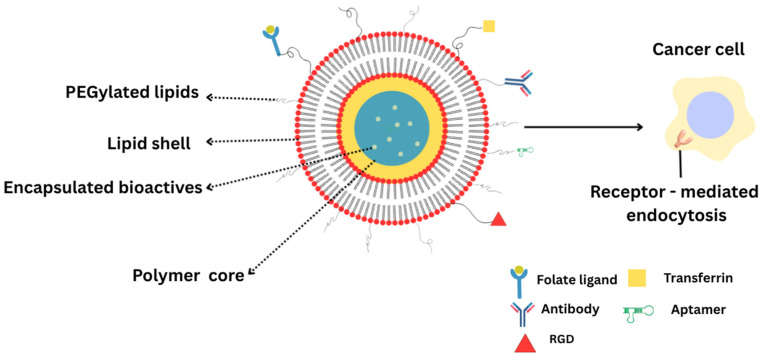
Schematic view of functionalized LPHNPs showing ligand-conjugated lipid shells enabling receptor-mediated targeting and enhanced intracellular drug delivery. RGD: arginine-glycine-aspartic acid.

**Figure 3 pharmaceuticals-18-01772-f003:**
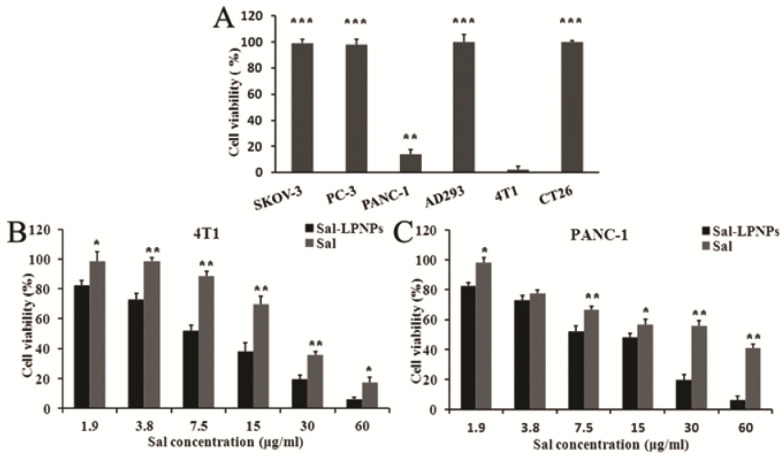
Sal-LPNPs and free Sal showing cytotoxic effects. (**A**) Three human cancer cell lines (SKOV-3, PC-3, and PANC-1), two mouse cancer cell lines (CT26 and 4T1), and one normal human cell line (AD293) were treated with 100 μg/mL Sal for 48 h. (**B**) Sal-LPNPs and free Sal’s cytotoxic effects on 4T1 cells after 48 h. (**C**) Sal-LPNPs and free Sal’s cytotoxic effects on PANC-1 cells after 48 h. * *p* < 0.05 in comparison to 4T1. When compared to free Sal, Sal-LPNPs showed noticeably greater, dose-dependent anticancer efficacy (** *p* < 0.01, *** *p* < 0.001). This figure is adapted from [129], used under a CC BY 4.0 license.

**Figure 4 pharmaceuticals-18-01772-f004:**
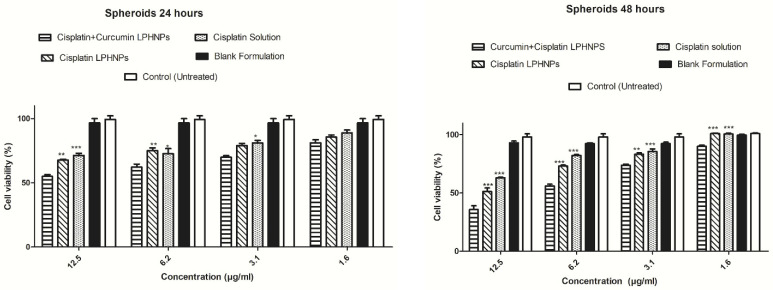
Results of cytotoxicity studies carried out on 3D spheroids treated with curcumin and cisplatin co-loaded LPHNPs, cisplatin-loaded LPHNPs, and cisplatin solution. Values represent mean ± SD (n = 3) (* *p* < 0.05, ** *p* < 0.01, *** *p* < 0.001). This figure is adapted from Dove Medical Press Ltd., Reference [136], used under a CC BY 3.0 license.

**Figure 5 pharmaceuticals-18-01772-f005:**
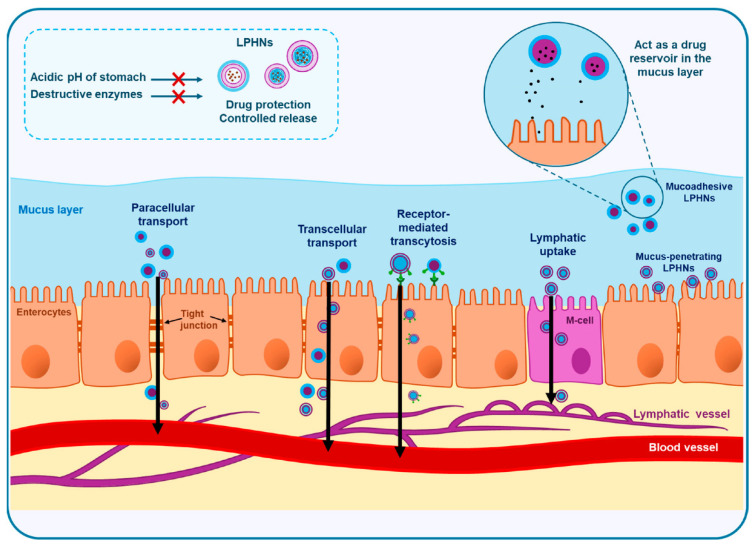
Schematic diagram illustrating the major intestinal absorption mechanisms of LPHNPs. This figure is adapted from [26], used under a CC BY 4.0 license.

**Figure 6 pharmaceuticals-18-01772-f006:**
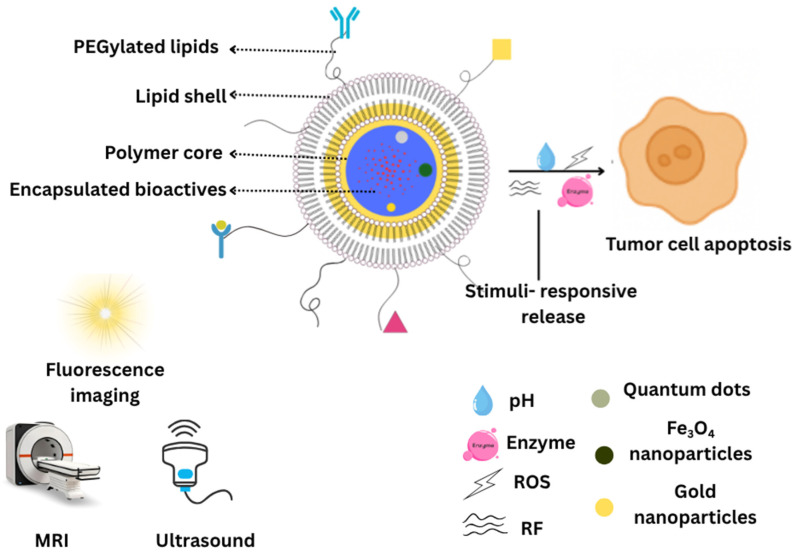
Schematic representation of theranostic LPHNPs showing integrated therapeutic and imaging components enabling simultaneous drug delivery, real-time tracking, and stimuli-responsive controlled release. MRI: magnetic resonance imaging, ROS: reactive oxygen species, RF: radio frequency.

**Table 1 pharmaceuticals-18-01772-t001:** Summary of nanocarrier types, their sizes, properties, advantages, limitations, examples, and clinical status.

Nanocarrier	Representative Particle Size/Typical Drug Loading	Typical Characteristics	Key Advantages	Major Limitations	Representative Examples	Clinical Stages	References
Vesicular carriers	50–300 nm/1–10%	Bilayered vesicles (e.g., liposomes, niosomes) with aqueous core	Biocompatible, encapsulates hydrophilic and lipophilic drugs, surface-modifiable	Physical/chemical instability, short circulation time, high production cost	Liposomes, niosomes, ethosomes, transfersomes, transethosomes, invasomes, cubosomes	Approved clinical trials	[10,11,12,13,14]
Polymeric nanoparticles	100–1000 nm/5–20%	Biodegradable polymer-based solid particles (100–1000 nm)	Intracellular release, controlled drug release, high stability, scalable production	Low cell interaction, poor delivery efficiency, potential for burst release, toxic monomer accumulation risks, scalability issues	Polylactic-co-glycolic acid (PLGA) nanoparticles, chitosan nanoparticles	Approved formulations and clinical research	[15]
Inorganic nanoparticles	10–100 nm/variable	Metal/metal oxide-based particles (e.g., gold, silica, iron oxide)	High imaging and diagnostic potential, magnetic/photothermal properties	Cytotoxicity, long-term biocompatibility concerns	Gold nanoparticles, iron oxide nanoparticles, silica nanoparticles	Preclinical and some clinical investigation	[16,17]
Polymeric micelles	20–100 nm/1–5%	Self-assembled amphiphilic block copolymers forming core–shell structures	Solubilize hydrophobic drugs, passive and active targeting possible	Low drug loading for hydrophilic drugs, instability in circulation	Polyethylene glycol (PEG)-polylactic acid (PLA) micelles, PCL-based micelles	FDA-approved clinical trials	[18]
Dendrimers	<10 nm/high	Highly branched 3D polymers with tunable surface groups	High drug loading, functionalizable, penetrates biological barriers	Complex synthesis, potential toxicity, limited clinical use	PAMAM dendrimers, polylysine dendrimers	Preclinical and early clinical	[19]
Carbon nanotubes/graphene	1–50 nm diameter/variable	One-dimensional carbon structures with high surface area	Exceptional mechanical strength, cellular uptake, potential for gene/drug delivery	Toxicity concerns, difficulty in excretion, regulatory challenges	Single-walled carbon nanotubes, graphene oxide	Preclinical	[20]
Quantum dots	2–10 nm/variable	Semiconductor nanocrystals with size-dependent fluorescence	Excellent optical properties for imaging and tracking	Heavy metal content, photobleaching, toxicity concerns	CdSe/ZnS quantum dots, InP quantum dots	Preclinical; limited clinical use	[21]
Lipid-based nanosystems	50–300 nm/1–5%	Solid lipid nanoparticles, nanostructured lipid carriers	Biocompatible, suitable for both hydrophilic and hydrophobic drugs	Low drug loading in the lipid matrix, lipid oxidation risk, drug expulsion during storage, tendency to aggregate, high instability in biological fluids	Solid lipid nanoparticles, nanostructured lipid carriers	Some marketed, ongoing clinical trials	[22]
Nanosuspensions	200–1000 nm/nearly 100% drug	Surfactant-stabilized colloidal dispersions of pure drug particles, usually with a particle size of less than 1 µm	Improves solubility and dissolution of poorly soluble drugs, suitable for parenteral and oral delivery	Physical instability (aggregation, sedimentation), high energy input required	Nanocrystals	Several approved formulations	[23]
Nanoemulsions	20–200 nm/low	Thermodynamically unstable systems of two immiscible liquids stabilized by surfactants; droplet size typically 20–200 nm	Enhances solubility and absorption of hydrophobic drugs, suitable for oral, topical, and parenteral routes	Thermodynamic instability, sensitivity to environmental conditions, limited drug loading for hydrophilic drugs	Microemulsions (o/w or w/o)	Approved products and clinical trials	[24]
Lipid-polymer hybrid nanoparticles	50–200 nm/5–15%	Core–shell nanostructures comprising a biodegradable polymeric core surrounded by a lipid layer	High stability and controlled release; enhanced cellular uptake and pharmacokinetics; potential for co-delivery of hydrophilic and hydrophobic drugs	Complex formulation and scale-up; possible phase separation, stability challenges under physiological conditions.	PEG-PLGA/1,2-Distearoyl-sn-glycero-3-phosphoethanolamine (DSPE) hybrid systems	Preclinical and clinical research	[25]

**Table 2 pharmaceuticals-18-01772-t002:** Overview of key material categories used in LPNPs, including representative examples, approved or clinical-stage products, functional roles, major advantages and disadvantages, and typical therapeutic applications.

Category	Examples	Approved Drug Products	Role	Advantages	Disadvantages	Applications
Lipids	Phosphatidylcholine, cholesterol, 1,2-distearoyl-sn-glycero-3-phosphocholine, 1,2-dioleoyl-sn-glycero-3-phosphoethanolamine	Doxil^®^ (liposomal doxorubicin), AmBisome^®^ (liposomal amphotericin B)	Form lipid shell; enhance stability; improve drug encapsulation	Biocompatible, flexible bilayer formation supports hydrophobic and hydrophilic drugs	May require stabilizers; prone to oxidation and hydrolysis	Drug delivery systems, sustained release formulations
Polymers	Polylactic-co-glycolic acid, PLA, polycaprolactone, chitosan, PEG-modified polymers	Lupron Depot^®^ (PLGA-based), Somatuline^®^ depot	Provide structural core; control drug release; ensure stability	Biodegradable, tunable degradation rate, mechanical strength	May induce burst release; hydrophobic core limits hydrophilic drug loading	Cancer therapy, protein/peptide delivery, vaccines
PEGylated lipids	DSPE-N-[methoxy(PEG)], 1,2-dimyristoyl-sn-glycero-3-phosphoethanolamine-N-[methoxy(PEG)], 1,2-dioleoyl-sn-glycero-3-phosphoethanolamine-N-[methoxy(PEG)]	Comirnaty^®^ (Pfizer–BioNTech mRNA vaccine), Spikevax^®^ (Moderna mRNA vaccine)	Provide stealth properties; reduce opsonization; extend circulation time	Increases half-life, prevents aggregation, improves biodistribution	May reduce cellular uptake (PEG dilemma)	Long-circulating drug carriers, tumor targeting
Charged lipids	1,2-Dioleoyl-3-trimethylammonium-propane, dioleoyl phosphatidic acid	Approved mRNA vaccines (Comirnaty^®^, Spikevax^®^)	Facilitate electrostatic interaction with nucleic acids; assist self-assembly	Enhance nucleic acid loading, promote endosomal escape	Can be cytotoxic at high concentrations	Gene delivery, mRNA vaccines
Surfactants/stabilizers	Poloxamer 188, poloxamer 407, polysorbate 80	Taxol^®^ (contains Polysorbate 80), Taxotere^®^	Prevent aggregation; improve dispersion; stabilize nanoparticles	Enhance solubility, reproducibility in preparation	Possible hypersensitivity reactions; surfactant residues	Emulsification, nanoprecipitation, drug solubilization
Targeting ligand conjugates	Folic acid- PEG, arginine-glycine-aspartic acid peptide-PEG, antibody-PEG conjugates	Mylotarg^®^ (CD33-targeted ADC), Enhertu^®^ (HER2-targeted ADC)	Enable active targeting to receptors; improve cellular uptake	Increased specificity, reduced off-target effects	Complex synthesis; high cost	Tumor-targeted therapy, receptor-mediated delivery

**Table 3 pharmaceuticals-18-01772-t003:** Overview of phytochemicals delivered using LPNPs, detailing the preparation methods, lipid-polymer compositions, therapeutic applications, in vivo models, routes of administration, comparative controls, and key observations.

Name	Preparation Method	Composition (Lipid/Polymer)	Medical Conditions	Therapeutic Application	In Vivo Model	Route of Administration	Comparative Control	Key Observations	Reference
Ursolic acid	Nanoprecipitation	PLGA Resomer RG 503 H, PLGA, soy phosphatidylcholine, phospholipon 90G, dimethyldioctadecyl-ammonium (bromide salt), DSPE-PEG 2000	Oncological	Pancreatic ductal adenocarcinoma	Xenograft mouse model (AsPC-1, BxPC-3)	Intravenous	Free ursolic acid	Nanocarriers demonstrated excellent physicochemical and biological characteristics-IC_50_ below 20 µM, particle size around 150 nm, encapsulation efficiency up to 70%, and high stability. Cytotoxicity assays on AsPC-1 and BxPC-3 cells, hemolysis testing, and TEM imaging confirmed their activity and safety.	[34]
Thymoquinone	Single-step nanoprecipitation method	Chitosan, phospholipon 90G	Oncological	Breast cancer	Tumor-bearing mice; oral pharmacokinetic model	Oral	Free thymoquinone	Optimized thymoquinone-loaded nanoparicles exhibited favorable properties, including particle size < 200 nm, polydispersity index (PDI) < 0.25, entrapment efficiency > 85%, and zeta potential > 25 mV. They demonstrated strong stability, sustained drug release for up to 48 h, and high mucin-binding efficiency (>70%). In vitro and ex vivo studies showed significantly improved anti-breast cancer activity in MDA-MB-231 and MCF-7 cell lines, intestinal permeation, and oral bioavailability (4.74-fold higher) compared to free phytochemical.	[121]
Emodin	Nanoprecipitation method	PLGA copolymer, soybean lecithin, DSPE-PEG 2000	Oncological	Breast cancer	Breast tumor xenograft mouse model	Intravenous	Free emodin	Nanoparticles had an average particle size of 122.7 ± 1.79 nm and entrapment efficiency of 72.8%. In comparison to free emodin, emodin-loaded nanoparticles showed enhanced cytotoxicity against MCF-7 breast cancer cells by increasing drug uptake and promoting early apoptosis. The elevated Bax/Bcl-2 ratio confirmed apoptosis induction as the primary anticancer mechanism. In vivo, emodin-nanoparticles inhibited tumor growth by over 60%, likely due to improved passive targeting at the tumor site.	[122]
Abietic acid	Microinjection technique	Chitosan, cholesterol	Non-oncological	Antioxidant and anti-inflammatory	Rodent inflammatory model; ex vivo gut permeation	Oral	Pure abietic acid	Optimized nanoparticles demonstrated particle size of 384.5 ± 6.36 nm, PDI of 0.376, zeta potential of 23.0 mV, and encapsulation efficiency of 80.01 ± 1.89%. Hybrid nanoparticles enhanced ex vivo gut permeation by 2.49-fold, attributed to lipid and surfactant components. The formulation showed markedly higher antioxidant and anti-inflammatory activities (21.51 ± 2.23% swelling vs. 46.51 ± 1.74% for pure phytochemical.	[70]
*β*-Sitosterol	Single-step nanoprecipitation method	PLGA Resomer RG 503 H, PLGA, DSPE-PEG 2000	Non-oncological	Hepatoprotective	Carbon tetrachloride induced hepatotoxicity rat model	Oral	Free β-Sitosterol	In a CCl_4_-induced hepatotoxicity rat model, β-Sitosterol-LPHNPs (400 mg/kg) effectively normalized ALT, AST, MDA, CAT, bilirubin, and albumin levels without inhibiting CYP2E1 activity. Histological and immunohistochemical analyses confirmed preservation of normal liver architecture and reduced cleaved caspase-3 expression, indicating a strong hepatoprotective effect of the formulation.	[123]
Tartary buckwheat (TBFs) extracts	Single-step solvent evaporation with ultrasound	DSPE-N-[methoxy PEG-2000]), poly(D,L-lactide-co-glycolide, MW = 5000, lactide: glycolide ratio 50:50), egg lecithin, cholesterol	Non-oncological	Immuno-modulator	Immuno-suppressed mouse model	Oral	Free TBFs extract	Optimized formulation demonstrated high encapsulation efficiency (96.4 ± 1.1%), uniform nanosize (61.25 ± 1.83 nm), spherical morphology, and excellent stability. Compared to free TBFs, TBFs/LPHNPs showed stronger antioxidant and anti-inflammatory activities in RAW 264.7 macrophages and improved intestinal absorption through enhanced Caco-2 transmembrane transport. In vivo studies further confirmed an enhanced immune response in immunosuppressed mice.	[124]

**Table 4 pharmaceuticals-18-01772-t004:** Quantitative assessment of therapeutic performance across LPHNP systems.

Drug/Phytochemical	LPHNP Composition	Application and Model	Quantitative Outcomes	Reference
Curcumin + Paclitaxel	PLGA-core/lipid shell, chitosan-coated	Breast cancer (in vitro); pharmacokinetic evaluation in rats	IC_50_ reduced from 480.06 to 282.97 µg/mL; AUC increased 3.8-fold (CUR) increased 6.6-fold (paclitaxel)	[134]
Salidroside	PLGA-PEG-PLGA, lecithin/cholesterol	Pancreatic cancer (in vitro); PANC-1, 4T1 cells	IC_50_: 9.54 (PANC-1), 8.23 (4T1) µg/mL	[129]
Chrysin + Piperine	Chitosan-lecithin hybrid	Pancreatic cancer (in vitro); PANC cells; HFF normal cells	IC_50_: 14 µg/mL (PANC); >500 µg/mL (HFF)	[142]
Docetaxel + Curcumin	Polymer-lipid hybrid	Prostate cancer (in vitro); PC-3 prostate tumor xenograft	Significant tumor inhibition; docetaxel-CUR-LPHNPs group exhibited the highest tumor inhibition rate (82.5%), followed by docetaxel-CUR-nanoparticles (62.1%) and docetaxel-LPHNPs (45.2%).	[137]
Cisplatin + Curcumin	Hybrid dual-delivery	Cervical cancer (in vitro); HeLa/HUVEC; cervical tumor mice	Synergistic cytotoxicity; superior in vivo tumor suppression when compared with coloaded polymeric nanoparticles	[138]
Hydroxycamptothecin	LPHNPs via modified solvent evaporation method utilizing PLGA, DSPE-PEG_2000_ and lecithin	Breast cancer (in vitro) MCF-7 cells; liver carcinoma (in vitro) HepG2 cells; in vivo pharmacokinetics	LPNPs exhibited lower IC_50_ and reduced cell viability; 3× higher bioavailability compared to drug solution in rats	[32]
Enoxaparin	Oral chitosan–lipid hybrid nanoparticles via self-assembly method using glyceryl monooleate	Bioavailability enhancement via in vivo anticoagulant activity in rats	Bioavailability increased 5-fold compared to enoxaparin solution. Nanoparticles with a glyceryl monooleate/chitosan ratio of 0.2 showed oral bioavailability ~10%	[35]
Factor VII siRNA or luciferase-encoding mRNA	Parallelized microfluidic device	Gene silencing in mice	A 4-fold increase in hepatic gene silencing and a 5-fold enhancement in luciferase expression	[176]

**Table 5 pharmaceuticals-18-01772-t005:** Patent applications (https://www.wipo.int/portal/; accessed on 30 October 2025) related to lipid-polymer hybrid nanoparticles in the last five years.

Application ID	Publication Date	Title	Patent Status	Summary of Invention
202021056479	25 December 2020	Design and development of lipid-polymer hybrid nanoparticles for combinatorial drug delivery	Pending	The invention describes the development of lipid-polymer hybrid nanoparticles that address challenges in cancer therapy by offering stability, biocompatibility, and tunable surface properties for targeted delivery and controlled release.
17228224	12 April 2021	Dual-targeting lipid-polymer hybrid nanoparticles	Pending	The invention discloses dual-targeting polymer-lipid hybrid nanoparticles made of a lipid shell functionalized with a targeting moiety and a polymeric core containing a heme oxygenase-1 inhibitor.
2021101545	26 March 2021	Method for formation of lipid-polymer hybrid nanoparticles for combinatorial vincristine sulfate and lomustine drug delivery	Granted	The disclosure describes lipid-polymer hybrid nanoparticles capable of co-encapsulating vincristine sulfate and lomustine, with tunable surface properties for targeted delivery and controlled release.
202211651315.3	22 December 2022	Application of polymer lipid hybrid nanoparticles as immunologic adjuvant and immune preparation	Granted	Formulation composed of biodegradable amphiphilic block copolymers and lipids. When combined with immunopotentiators, nanoparticles significantly enhance humoral and cellular immune responses.
23710967	23 February 2023	Polymer-lipid hybrid nanoparticles comprising a lipid and a block copolymer as well as methods of making and uses thereof	Pending	Formulation designed to encapsulate protein or polynucleotide antigens, making them useful as vaccines, pharmaceuticals, targeted delivery systems, and non-viral nucleotide carriers.
202310252352.5	15 March 2023	Application of emodin polymer lipid hybrid nanoparticles	Pending	Formulation to improve breast cancer therapy by targeting tumor sites and releasing emodin to inhibit the IL-6/JAK2/STAT3 pathway, thereby overcoming drug resistance.
202341017235	15 March 2023	Polymeric lipid hybrid nanoparticles for controlled release system by the nanoprecipitation method	Pending	Formulation prepared using PLGA as the core and lecithin-PEG 2000 as the lipid shell for myocardial infarction therapy to improve amlodipine’s pharmacokinetics and solubility.
202410758382.8	13 June 2024	Polymer for mRNA delivery, lipid/polymer hybrid nanoparticle using same, and preparation method and application thereof	Granted	A cationic poly(β-amino ester) polymer is developed for effective mRNA delivery, formulated into lipid-polymer hybrid nanoparticles with auxiliary lipids like DOTAP.
WO/2024/252406	12 December 2024	Lipid-polymer hybrid nanoparticles	Pending	Formulation consisting of a biodegradable polymeric core and a lipid shell, where most of the drug adheres to the inner lipid layer.

## Data Availability

No new data were created or analyzed in this study. Data sharing is not applicable to this article.

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
