# Peer review of "Advances in Lipid-Polymer Hybrid Nanoparticles: Design Strategies, Functionalization, Oncological and Non-Oncological Clinical Prospects"

_pharmaceuticals, 2025, doi:10.3390/ph18121772_

Round 1

Reviewer 1 Report

Comments and Suggestions for Authors

This manuscript on Lipid-Polymer Hybrid Nanoparticles (LPHNPs) provides a comprehensive, well-structured, and timely review of the latest advances in this rapidly evolving field, covering design strategies, functionalization, and clinical prospects. It possesses significant scholarly value and publication potential. However, there is room for improvement in the depth of certain sections, the intuitiveness of illustrations, the timeliness of data updates, and linguistic precision. A revision is recommended prior to acceptance.

  1. It is highly recommended to add high-quality schematic illustrations in key sections such as "Functionalization Strategies" and "Theranostics" to visually demonstrate the targeting mechanisms and theranostic principles of LPHNPs, thereby enhancing the manuscript's readability and impact.
  2. The transitions between sections should be reviewed to ensure a smoother narrative flow. For instance, the connection between the description of various structures and their corresponding preparation methods could be more seamlessly articulated.
  3. The "Challenges and Future Perspectives" section should provide a more concrete discussion on how to address scale-up manufacturing challenges through "Quality-by-Design (QbD)" and advanced manufacturing technologies, moving beyond merely mentioning the concepts.
  4. The "Patents and Clinical Trials" section must explicitly state the current absence of clinical trials for canonical LPHNPs, despite existing patents. It is advised to strengthen this discussion by analyzing the underlying reasons (e.g., regulatory, manufacturing hurdles) and to provide a more proactive outlook on the future pathway to clinical application.
  5. Numerical data, such as percentages and fold-increases (e.g., "5-fold chancement" should be corrected to "5-fold enhancement"), must be carefully checked and standardized throughout the manuscript for accuracy and consistency.
  6. It is suggested to refer to the latest research progress on Lipid-polymer hybrid nanoparticles for writing and discussion, such as   https://doi.org/10.1002/EXP.20210217;   https://doi.org/10.1002/EXP.20230137

Author Response

Reviewer 1

Comments and Suggestions for Authors

This manuscript on Lipid-Polymer Hybrid Nanoparticles (LPHNPs) provides a comprehensive, well-structured, and timely review of the latest advances in this rapidly evolving field, covering design strategies, functionalization, and clinical prospects. It possesses significant scholarly value and publication potential. However, there is room for improvement in the depth of certain sections, the intuitiveness of illustrations, the timeliness of data updates, and linguistic precision. A revision is recommended prior to acceptance.

Reply: We are happy to see the positive comments from the eminent reviewer. We would also like to thank you for raising the concerns, which would improve the quality of our manuscript.

  1. It is highly recommended to add high-quality schematic illustrations in key sections such as "Functionalization Strategies" and "Theranostics" to visually demonstrate the targeting mechanisms and theranostic principles of LPHNPs, thereby enhancing the manuscript's readability and impact.

Reply: We thank the reviewer for these suggestions. We have added 2 new figures as suggested by the eminent reviewer in the revised text (Figures 2 and 6).

  1. The transitions between sections should be reviewed to ensure a smoother narrative flow. For instance, the connection between the description of various structures and their corresponding preparation methods could be more seamlessly articulated.

Reply: We thank the reviewer for this observation. We have added connection sentences at the end of the structural description section and at the beginning of the preparation methods as suggested by the eminent reviewer (Section 2.1.3 and Section 2.3).

  1. The "Challenges and Future Perspectives" section should provide a more concrete discussion on how to address scale-up manufacturing challenges through "Quality-by-Design (QbD)" and advanced manufacturing technologies, moving beyond merely mentioning the concepts.

Reply: We thank the reviewer for this suggestion. Based on another reviewer's comment, a new paragraph (Section 2.4) on Quality by Design (QbD) considerations in LPHNP development has been added to the revised manuscript, outlining how the integration of QbD principles enhances scientific understanding, improves scalability, and aligns formulation practices with regulatory expectations for advanced nanomedicine products.

  1. The "Patents and Clinical Trials" section must explicitly state the current absence of clinical trials for canonical LPHNPs, despite existing patents. It is advised to strengthen this discussion by analyzing the underlying reasons (e.g., regulatory, manufacturing hurdles) and to provide a more proactive outlook on the future pathway to clinical application.

Reply: We thank the reviewer for this suggestion. We have added a new paragraph (in section 5.1) to strengthen the discussion on the future pathway to clinical application in the revised text.

  1. Numerical data, such as percentages and fold-increases (e.g., "5-fold chancement" should be corrected to "5-fold enhancement"), must be carefully checked and standardized throughout the manuscript for accuracy and consistency.

Reply: We are sorry for this typo error. We carefully reviewed and corrected the numerical values to maintain consistency throughout the manuscript.

  1. It is suggested to refer to the latest research progress on Lipid-polymer hybrid nanoparticles for writing and discussion, such as   https://doi.org/10.1002/EXP.20210217;   https://doi.org/10.1002/EXP.20230137

Reply: We thank the reviewer for this suggestion. We have incorporated the information on both articles in the revised article (introduction and section 4.1.3).

Reviewer 2 Report

Comments and Suggestions for Authors
  • In abstract, the phrase “new generation nanocarriers” is repetitive and somewhat promotional; consider replacing it with “next-generation” or simply “hybrid nanocarriers.”
  • The section on patents could benefit from specifying the time frame or number of patents analyzed.
  • In the introduction, the phrase “new generation platform” could be replaced with “next-generation nanocarrier system” for formality.
  • When referring to mechanisms like the “enhanced permeability and retention effect,” consider abbreviating it as “EPR effect” after first mention.
  • Repetition of phrases like “improve bioavailability” and “enhance therapeutic efficacy” could be reduced for conciseness.
  • For Table 1, Suggestion: Add a column or footnote summarizing translational readiness or clinical stage (e.g., “Approved,” “Preclinical,” “Under investigation”) to provide context and interpretive value.
  • Alternate use of “LPHNs” and “LPHNPs” throughout; standardize to one acronym.
  • Chemical names: When first mentioned, ensure standard IUPAC capitalization (e.g., “1,2-distearoyl-sn-glycero-3-phosphoethanolamine”).
  • Define key terms (DSPE, PEG) at first use only, then use abbreviations consistently thereafter.
  • Superficial Discussion of Material–Property Relationships. The section lists materials (e.g., PLGA, phosphatidylcholine, DSPE-PEG) and their functions, but does not sufficiently connect their physicochemical characteristics (e.g., hydrophobicity, molecular weight, crystallinity) with resulting performance outcomes (e.g., release kinetics, biodistribution).
  • Include a brief table or sentence summarizing quantitative advantages (e.g., Microfluidic systems typically reduce PDI from 0.25 to <0.1 and improve EE by 10–15% compared to bulk mixing.
  • Provide units consistently (e.g., µg/mL, nm) with proper formatting
  • Replace informal phrasing (“promising results noticed in various studies”) with academic alternatives (“multiple reports have demonstrated)
  • Explain why certain methods (e.g., microfluidics) achieve better siRNA encapsulation or uniformity, or what trade-offs exist (e.g., scalability vs. complexity).
  • Typographical errors: “fabricated LPHNPs achieved a high siRNA encapsulation (~80%)” → “The fabricated LPHNPs achieved high siRNA encapsulation (~80%).”
  • Avoid abbreviation overload in one sentence (e.g., “AUC,” “t½,” “TNBC,” “EGFR,” “TROP2”)—expand at first mention or provide brief parenthetical definitions for accessibility.
  • The phrase “yielding size-controlled, siRNA-loaded LPHNPs with high encapsulation efficiency and favorable safety” is vague. Quantify “favorable safety” or provide reference to toxicity metrics.
  • Functionalization is discussed superficially (e.g., “ligands enhance tumor specificity”), without mechanistic detail on ligand–receptor kinetics, intracellular trafficking, or receptor recycling effects on delivery efficiency.
  • Table 4 is overloaded please try to decrease its content as much as possible

Author Response

Reviewer 2

Comments and Suggestions for Authors

Reply: We would like to convey our sincere thanks for the time and suggestions from the potential reviewer, which would surely improve the quality of the article.

  • In abstract, the phrase “new generation nanocarriers” is repetitive and somewhat promotional; consider replacing it with “next-generation” or simply “hybrid nanocarriers.”

Reply: We thank the reviewer for this suggestion. We have changed it accordingly in the abstract as suggested by the eminent reviewer.

  • The section on patents could benefit from specifying the time frame or number of patents analyzed.

Reply: We thank the reviewer for this comment. The recent patent for the last 5 years (2020-2024) was included in the text. This information is now included in the revised text (section 5.1).

  • In the introduction, the phrase “new generation platform” could be replaced with “next-generation nanocarrier system” for formality.

Reply: We thank the reviewer for this suggestion. We have changed it accordingly in the introduction section as suggested by the eminent reviewer.

  • When referring to mechanisms like the “enhanced permeability and retention effect,” consider abbreviating it as “EPR effect” after first mention.

Reply: We agreed and added an abbreviation for “EPR effect” after it was first mentioned.

  • Repetition of phrases like “improve bioavailability” and “enhance therapeutic efficacy” could be reduced for conciseness.

Reply: We thank the reviewer for this comment. The whole manuscript has been revised to minimize the repetition of the said terminologies and replaced with concise and varied wordings.

  • For Table 1, Suggestion: Add a column or footnote summarizing translational readiness or clinical stage (e.g., “Approved,” “Preclinical,” “Under investigation”) to provide context and interpretive value.

Reply: We agreed and added in Table as suggested by the reviewer (Table 1).

  • Alternate use of “LPHNs” and “LPHNPs” throughout; standardize to one acronym.

Reply: We appreciate the keen observation of the reviewer. The content has been carefully revised to use the acronym “LPHNPs” consistently throughout the manuscript.

  • Chemical names: When first mentioned, ensure standard IUPAC capitalization (e.g., “1,2-distearoyl-sn-glycero-3-phosphoethanolamine”).

Reply: We thank the reviewer for this suggestion. We have checked all chemical names and added abbreviations according to IUPAC nomenclature.

  • Define key terms (DSPE, PEG) at first use only, then use abbreviations consistently thereafter.

Reply: We thank the reviewer for this suggestion. We have revised the whole manuscript and included the description and abbreviations accordingly.

  • Superficial Discussion of Material–Property Relationships. The section lists materials (e.g., PLGA, phosphatidylcholine, DSPE-PEG) and their functions, but does not sufficiently connect their physicochemical characteristics (e.g., hydrophobicity, molecular weight, crystallinity) with resulting performance outcomes (e.g., release kinetics, biodistribution).

Reply: We thank the reviewer for the comment. The connection between the physicochemical properties of the materials (e.g., hydrophobicity, molecular weight, crystallinity) and their performance outcomes (e.g., release kinetics, biodistribution) has now been clearly elaborated in the materials section of the revised manuscript.

  • Include a brief table or sentence summarizing quantitative advantages (e.g., Microfluidic systems typically reduce PDI from 0.25 to <0.1 and improve EE by 10–15% compared to bulk mixing.

Reply: We thank the reviewer for this suggestion. A separate pargarph is now included in the revised text accordingly (section 2.3.3).

  • Provide units consistently (e.g., µg/mL, nm) with proper formatting

Reply: Agreed and modified accordingly throughout the manuscript.

  • Replace informal phrasing (“promising results noticed in various studies”) with academic alternatives (“multiple reports have demonstrated)

Reply: We agreed and modified accordingly in the revised text (Section 4.1.1).

  • Explain why certain methods (e.g., microfluidics) achieve better siRNA encapsulation or uniformity, or what trade-offs exist (e.g., scalability vs. complexity).

Reply: Thank you for the comment. Additional details have been added under the microfluidic technique section.

  • Typographical errors: “fabricated LPHNPs achieved a high siRNA encapsulation (~80%)” → “The fabricated LPHNPs achieved high siRNA encapsulation (~80%).”

Reply: We agreed and modified accordingly in the revised text (Section 4.1.3).

  • Avoid abbreviation overload in one sentence (e.g., “AUC,” “t½,” “TNBC,” “EGFR,” “TROP2”)—expand at first mention or provide brief parenthetical definitions for accessibility.

Reply: We agreed and expanded all the abbreviations in the revised text (Section 4.1.3).

  • The phrase “yielding size-controlled, siRNA-loaded LPHNPs with high encapsulation efficiency and favorable safety” is vague. Quantify “favorable safety” or provide reference to toxicity metrics.

Reply: We are sorry for this error. The said article did not have information regarding the quantity or toxic metrics. Hence, the terminology “favorable safety” has been removed from the revised text.

  • Functionalization is discussed superficially (e.g., “ligands enhance tumor specificity”), without mechanistic detail on ligand–receptor kinetics, intracellular trafficking, or receptor recycling effects on delivery efficiency.

Reply: hank you for the comment. Additional mechanistic details have been added under functionalization strategies for targeted therapy, addressing ligand–receptor kinetics, intracellular trafficking, and receptor recycling effects on delivery efficiency.

  • Table 4 is overloaded please try to decrease its content as much as possible

Reply: We thank the reviewer for this suggestion. The table has been thoroughly revised, and the content has been made more concise in the updated version.

Reviewer 3 Report

Comments and Suggestions for Authors

The present manuscript entitled “Advances in Lipid-Polymer Hybrid Nanoparticles: Design Strategies, Functionalization, and Clinical Prospects” presents a comprehensive review on lipid-polymer hybrid nanoparticles (LPHNPs) demonstrates substantial breadth in covering design strategies, functionalization approaches, and therapeutic applications. However, several significant issues related to novelty, presentation quality, and methodological rigor require attention before publication.

  1. The manuscript primarily compiles existing knowledge without providing substantial novel insights or critical perspectives. The review lacks comparative analysis between LPHNPs and other emerging hybrid systems. Critical evaluation of contradictory findings in the literature (e.g., the PEG dilemma mentioned in Table 2 but not adequately discussed) is missing. There is not gap analysis to identify the unexplored research areas. Incorporating systematic comparison tables with quantitative data (IC50 values, bioavailability enhancement factors, tumor inhibition percentages) across different LPHNP types and therapeutic applications would increase the depth and the presentation of the manuscript.
  2. The manuscript suffers from inconsistent organization which I observed in several sections. For eg. section 5 (Therapeutic Applications) is disproportionately long and could be divided into oncological and non-oncological applications. The transition from structural features (Section 2) to materials (Section 3) and then methods (Section 4) lacks logical flow. Theranostics (Section 7) appears disconnected from the main therapeutic applications discussion. I suggest to restructure it to group related content: (1) Design and Fabrication, (2) Functionalization Strategies, (3) Therapeutic Applications (subdivided by disease area), (4) Translation Challenges etc.
  3. The critical discussion about the limitations is insufficient. I suggest to expand section 10 with several subsections Batch-to-batch reproducibility, Storage stability, Immunotoxicity, Cost-effectiveness analysis, Patient stratification etc. Authors may consider including additional subsections based on their understanding and experience.
  4. Although there is separate section for clinical translation but the discussion is inadequate. The patent section (Section 9) acknowledges no registered clinical trials for canonical LPHNPs, yet the manuscript does not critically analyze why this translational gap exists despite extensive preclinical evidence.
  5. There is an inconsistent depth across the different applications. The manuscript provides extensive coverage of cancer applications but other areas viz. vaccine application, central nervous system targeting, antimicrobials is superficially treated. Either authors may consider including these areas in detail or consider including oncological applications somewhere in the title.
  6. Despite mentioning QbD in the abstract and conclusions, the manuscript does not systematically apply this framework. Authors may consider including a dedicated subsection on QbD principles for LPHNP development, including CQA identification and control strategies.
  7. Table 1: Should include representative particle sizes and typical drug loading capacities for each nanocarrier type. Table 2: Lacks specific examples of approved/clinical-stage products for each category. Table 3: Missing critical information on in vivo models used, routes of administration, and comparative controls. Table 4: Patent table should include patent status (granted vs. pending).

The manuscript requires substantial revisions addressing the major comments, particularly enhancing critical analysis, restructuring content organization, expanding clinical translation discussion, and improving data presentation consistency.

Author Response

Reviewer 3

Comments and Suggestions for Authors

The present manuscript entitled “Advances in Lipid-Polymer Hybrid Nanoparticles: Design Strategies, Functionalization, and Clinical Prospects” presents a comprehensive review on lipid-polymer hybrid nanoparticles (LPHNPs) demonstrates substantial breadth in covering design strategies, functionalization approaches, and therapeutic applications. However, several significant issues related to novelty, presentation quality, and methodological rigor require attention before publication.

Reply: We are happy to see the positive comments from the eminent reviewer. We would also like to thank you for raising the concerns, which would improve the quality of the manuscript.

  1. The manuscript primarily compiles existing knowledge without providing substantial novel insights or critical perspectives. The review lacks comparative analysis between LPHNPs and other emerging hybrid systems. Critical evaluation of contradictory findings in the literature (e.g., the PEG dilemma mentioned in Table 2 but not adequately discussed) is missing. There is not gap analysis to identify the unexplored research areas. Incorporating systematic comparison tables with quantitative data (IC50 values, bioavailability enhancement factors, tumor inhibition percentages) across different LPHNP types and therapeutic applications would increase the depth and the presentation of the manuscript.

Reply: We thank the reviewer for the valuable feedback. In response, the manuscript has been substantially revised to incorporate deeper critical perspectives, including a comparative analysis of LPHNPs with other emerging hybrid systems and an expanded discussion of contradictory findings such as the PEG dilemma. A dedicated gap analysis has been added to highlight unexplored research areas (Section 6). Furthermore, new systematic comparison table (Table 4) containing quantitative data (ICâ‚…â‚€ values, bioavailability enhancement, tumor inhibition percentages) have been included to strengthen the analytical depth and overall presentation of the review.

  1. The manuscript suffers from inconsistent organization which I observed in several sections. For eg. section 5 (Therapeutic Applications) is disproportionately long and could be divided into oncological and non-oncological applications. The transition from structural features (Section 2) to materials (Section 3) and then methods (Section 4) lacks logical flow. Theranostics (Section 7) appears disconnected from the main therapeutic applications discussion. I suggest to restructure it to group related content: (1) Design and Fabrication, (2) Functionalization Strategies, (3) Therapeutic Applications (subdivided by disease area), (4) Translation Challenges etc.

Reply: We thank the reviewer for these suggestions. We have thoroughly reorganized the manuscript by restructuring the content as suggested by the eminent reviewer.

  1. The critical discussion about the limitations is insufficient. I suggest to expand section 10 with several subsections Batch-to-batch reproducibility, Storage stability, Immunotoxicity, Cost-effectiveness analysis, Patient stratification etc. Authors may consider including additional subsections based on their understanding and experience.

Reply: We thank the reviewer for the constructive suggestion. Section 5.2 has been expanded and reorganized into subsections covering key limitations of LPHNPs, including batch-to-batch reproducibility, storage stability, immunotoxicity, cost-effectiveness, and patient stratification. This expanded section provides a more comprehensive and structured evaluation of the practical barriers affecting the clinical translation of LPHNPs.

  1. Although there is separate section for clinical translation but the discussion is inadequate. The patent section (Section 9) acknowledges no registered clinical trials for canonical LPHNPs, yet the manuscript does not critically analyze why this translational gap exists despite extensive preclinical evidence.

Reply: We thank the reviewer for this suggestion. A new paragraph has been added in section 5.1 to explain the factors contributing to the translational gap despite extensive preclinical evidence.

  1. There is an inconsistent depth across the different applications. The manuscript provides extensive coverage of cancer applications but other areas viz. vaccine application, central nervous system targeting, antimicrobials is superficially treated. Either authors may consider including these areas in detail or consider including oncological applications somewhere in the title.

Reply: We thank the reviewer for this comment. We acknowledge that the manuscript provides more extensive coverage of oncological applications, as this area represents the primary focus of our review; non-oncological applications have been included to provide broader context and completeness. In line with the reviewer’s recommendation, we have revised the title, abstract, and keywords to explicitly reflect the emphasis on oncological applications.

  1. Despite mentioning QbD in the abstract and conclusions, the manuscript does not systematically apply this framework. Authors may consider including a dedicated subsection on QbD principles for LPHNP development, including CQA identification and control strategies.

Reply: We thank the reviewer for this suggestion. A new paragraph (Section 2.4) on Quality by Design (QbD) considerations in LPHNP development has been added to the revised manuscript, outlining how the integration of QbD principles enhances scientific understanding, improves scalability, and aligns formulation practices with regulatory expectations for advanced nanomedicine products.

  1. Table 1: Should include representative particle sizes and typical drug loading capacities for each nanocarrier type. Table 2: Lacks specific examples of approved/clinical-stage products for each category. Table 3: Missing critical information on in vivo models used, routes of administration, and comparative controls. Table 4: Patent table should include patent status (granted vs. pending).

Reply: We thank the reviewer for these helpful suggestions regarding the completeness and clarity of the tables. All four tables have been revised accordingly.

The manuscript requires substantial revisions addressing the major comments, particularly enhancing critical analysis, restructuring content organization, expanding clinical translation discussion, and improving data presentation consistency.

Reply: We appreciate the time and effort put in by the eminent reviewer in reviewing our article. We have revised the whole manuscript, along with restructuring the content for better flow and added more description on translation challenges.

Round 2

Reviewer 2 Report

Comments and Suggestions for Authors

Accept in the present form

Reviewer 3 Report

Comments and Suggestions for Authors

The revisions are satisfactory and the revised manuscript can be accepted for publication.